# Agentic Surrogates: Automating Proxy Models of Simulators with Compute Aware Intelligence

## Abstract

Proxy (surrogate) models are indispensable for accelerating scientific computation, yet creating them remains a manual, sample-inefficient, and nonreproducible process, especially when simulators are costly and constrained by physics. We present a domain-agnostic controller that orchestrates domain-specific tools to eliminate human intervention in surrogate model construction while simultaneously achieving superior accuracy. Our system employs an intelligent controller that orchestrates every aspect of the surrogate creation process: it automatically determines where to sample next, when to switch acquisition strategies, which model architectures to deploy, and when the surrogate has reached sufficient quality. The controller treats different acquisition methods as a portfolio of experts and dynamically selects among them based on their actual performance in reducing error per unit of computational time. Crucially, the system adapts its modeling approach to the problem at hand, automatically deploying simpler models for linear relationships and sophisticated architectures for complex nonlinear behaviors. We establish theoretical guarantees for our adaptive acquisition strategy and prove bounds on sample complexity. Across diverse scientific computing benchmarks, our framework not only eliminates manual intervention but achieves 5.1% better final accuracy than the best hand-tuned approaches, while requiring 14.3% fewer simulator evaluations and 19.5% less wall-clock time. This represents a fundamental shift: surrogate modeling transforms from a labor-intensive craft requiring deep expertise into a push-button automated process that delivers superior results.

## 1 Introduction

Surrogate models have become essential for making high-fidelity simulators practical in real-world applications. Whether optimizing aircraft designs, tuning chemical processes, or exploring subsurface resources, engineers rely on these fast approximations to replace computationally expensive simulations. Yet despite their critical importance, creating accurate surrogate models remains a fundamentally manual process that can take weeks of expert effort and still produce suboptimal results.

The current state of practice requires engineers to make numerous interconnected decisions: how many initial samples to collect, where to sample next, which model architecture to use, when to switch strategies, how to balance exploration versus exploitation, and when to stop. Each decision affects all others, creating a complex optimization problem that practitioners navigate through intuition and trial-and-error. This manual process is not only time-consuming and expensive but also non-reproducible—different experts make different choices, leading to inconsistent results even on the same problem.

Prior work has explored using LLM agents for surrogate model automation in domain-specific contexts. For instance, a recent petroleum engineering study demonstrated that frontier LLMs could manage acquisition switching and achieve modest improvements over fixed strategies. However, this approach lacked formal guarantees and generalizability beyond its target domain. While these initial results were promising, they raised fundamental questions: Can acquisition switching be formalized with theoretical guarantees? How should computational cost factor into the decision process? Can the approach generalize across diverse simulators and physics constraints?

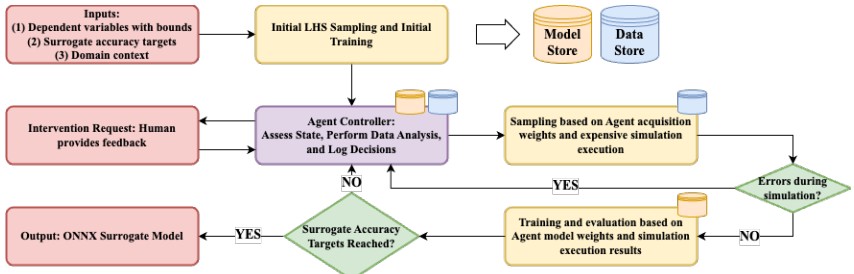

Figure 1: End-to-End Control Flow. The process begins with problem definition and initial Latin Hypercube Sampling (LHS). In the active loop, the Agent Controller assesses the surrogate's performance and assigns weights to the acquisition portfolio. Crucially, the Agent handles the Simulation Feedback Loop: if a simulation fails (e.g., non-convergence or physics violation), the Agent receives the error log, adjusts the search constraints, and retries, preventing wasted compute on invalid regions. The cycle continues until the dynamic stopping criteria (accuracy targets) are met or a human intervention is requested.

This work addresses these questions by presenting a comprehensive framework that automates surrogate construction with both theoretical foundations and practical effectiveness. We formalize the problem as cost-aware online learning over a portfolio of acquisition experts, introduce physics-informed models and stopping criteria, and demonstrate consistent improvements across diverse scientific computing applications. Our system not only eliminates manual intervention but achieves superior accuracy compared to expert-tuned baselines.

## 1.1 CONTRIBUTIONS

1. **Cost-Aware Problem Formulation**: We define surrogate model creation as a cost-minimization problem: achieve a target error while accounting for wall-clock latency and acquisition costs through an online portfolio of sampling strategies.

2. **Compute-Aware Adaptive Controller**: We design a compute-aware controller that adaptively combines residual-based, variance-based, Bayesian optimization–style, hybrid, and random acquisition strategies. The controller incorporates a multi-fidelity scheduler and physics-informed stopping rules.

3. **Multi-Model Architecture with Physics Integration**: We extend the framework to support multiple model classes (ANN, PINN, FNO), bias-corrected residual surrogates, heteroscedastic output heads, and correlation across multiple outputs.

4. **Comprehensive Benchmarks and Validation**: We introduce SimBench-Surrogate (well-network, gas processing plant, PDE proxy) and evaluate performance using calls-to-target, wall-clock time, calibration quality, and constraint violation rates, along with comprehensive ablations and reproducibility checks.

## 2 RELATED WORK

Our work builds upon advances in several areas: LLM-based automation, active learning for surrogates, and physics-informed modeling. We review the most relevant contributions and position our framework within this landscape.

### 2.1 LLM AGENTS FOR SCIENTIFIC AUTOMATION

Large language models have emerged as powerful orchestrators for complex scientific workflows. Plaat et al. (2025) provide a comprehensive survey of LLM-based autonomous agents, highlighting their ability to plan, reason, and adapt across diverse tasks. In the domain of data science, agents have evolved to handle increasingly complex pipelines. Hong et al. (2024) introduce Data Interpreter, an agent capable of solving long-term interconnected tasks and dynamic data adjustments through hierarchical graph modeling. Similarly, Zhang et al. (2025) present DeepAnalyze, an agent designed

for fully autonomous data science, from raw data to analyst-grade reports. Wang et al. (2025) further formalize this field by connecting general agent design principles with practical data science workflows, while Yano et al. (2025) demonstrate how LLMs can optimize post-training workflows.

While these systems excel at general data tasks, applying them to surrogate modeling for scientific simulators presents unique challenges. Previous works like Wuwu et al. (2025) have demonstrated automated PDE surrogation, but often focus on specific model types or lack the rigorous acquisition strategies needed for expensive black-box functions. Our framework addresses this by integrating the reasoning capabilities of these data science agents with a theoretically grounded, cost-aware acquisition portfolio specifically designed for scientific computing.

## 2.2 Acquisition Strategies and Active Learning

The choice of where to sample next fundamentally impacts surrogate quality and efficiency. Classical approaches include uncertainty-based sampling using Monte Carlo Dropout (Gal & Ghahramani, 2016), which provides principled uncertainty estimates for neural networks. Recent work has explored adaptive sampling strategies: Morales & Sheppard (2024) propose methods to reduce epistemic uncertainty, while Wang et al. (2024) introduce deep adaptive sampling that operates without labeled data.

Bayesian optimization provides another lens for acquisition design. Liu et al. (2024) enhance Bayesian optimization with LLMs to improve acquisition function selection, while Aglietti et al. (2025) use LLMs to generate novel acquisition functions through their FunBO framework. Zhang & Chen (2025) provide theoretical analysis of regret in Bayesian optimization settings. However, these approaches typically commit to a single acquisition strategy throughout the optimization process. Our work differs by maintaining a portfolio of acquisition experts and adaptively selecting among them based on time-normalized performance—an approach for which we provide formal regret guarantees.

## 2.3 Surrogate Modeling Frameworks and Tools

The surrogate modeling community has developed sophisticated toolboxes to support practitioners. Saves et al. (2023) present SMT 2.0, focusing on hierarchical and mixed variables, while Robani et al. (2025) extend this with explainability features. These tools provide essential building blocks but require manual orchestration and decision-making. Elman et al. (2025) demonstrate surrogate-based multilevel Monte Carlo for uncertainty quantification, highlighting the importance of calibrated uncertainty—a feature we incorporate through conformal prediction.

Physics-informed approaches have shown particular promise for scientific applications. Nadal et al. (2025) integrate PINNs into power system simulations, while Chandra et al. (2025) use Fourier Neural Operators for $CO_2$ storage decision-making, demonstrating the value of operator learning for PDE-based problems. Our framework uniquely combines multiple model classes (ANNs, PINNs, FNOs) and automatically selects among them based on problem characteristics and computational constraints.

## 2.4 Positioning Our Contribution

While prior work has made significant advances in individual components—LLM orchestration, acquisition strategies, or model architectures—no existing framework provides end-to-end automation with theoretical guarantees. Our key innovations relative to prior work include: (1) formalizing acquisition switching as online learning with proven regret bounds, (2) incorporating wall-clock time directly into the optimization objective, (3) automatically selecting and combining multiple model architectures based on problem structure, and (4) integrating physics constraints into both acquisition and stopping decisions. This comprehensive approach transforms surrogate modeling from a collection of tools requiring expert coordination into a fully automated, theoretically grounded system.

## 3 PROBLEM SETUP AND NOTATION

We consider the problem of automatically constructing surrogate models for expensive simulators while minimizing both computational cost and prediction error. This section formalizes the surrogate modeling task, defines our cost model, and introduces the portfolio-based acquisition framework.

### 3.1 PRELIMINARIES & NOTATION

We denote the input space by $\mathcal{X} \subseteq \mathbb{R}^d$ and the output space by $\mathcal{Y} \subseteq \mathbb{R}^m$. The target validation error is denoted by $\varepsilon$, and the total simulation budget is $B$.

### 3.2 SURROGATE MODELING TASK

Let $f^* : X \rightarrow Y$ denote a high-fidelity simulator mapping from a $d$-dimensional input space $X \subset \mathbb{R}^d$ to an $m$-dimensional output space $Y \subset \mathbb{R}^m$. In practice, we observe noisy evaluations:

$$y = f^*(x) + \xi, \quad \xi \sim \mathcal{N}(0, \Sigma(x)) \tag{1}$$

where $\xi$ represents potentially heteroscedastic noise. Our goal is to construct a surrogate model $\hat{f}_\theta : X \rightarrow Y$ with parameters $\theta$ that accurately approximates $f^*$ while minimizing the number of expensive simulator evaluations.

At iteration $t$, we maintain a dataset $\mathcal{D}_t = \{(x_i, y_i)\}_{i=1}^{n_t}$ of simulator evaluations, where $n_t$ denotes the total number of samples collected. The surrogate is trained to minimize a loss function $\mathcal{L}(\theta; \mathcal{D}_t)$, typically mean squared error for regression tasks.

### 3.3 COST MODEL

Each iteration of surrogate construction incurs three types of computational costs:

- **Acquisition cost** $\tau_t^{\text{acq}}$: Time to score and rank candidate points for sampling
- **Simulation cost** $\tau_t^{\text{sim}}$: Time to evaluate the simulator at selected points (possibly in parallel batches)
- **Training cost** $\tau_t^{\text{train}}$: Time to retrain or update the surrogate model

Our objective is to reach a target validation error $\mathcal{E}_V(\hat{f}_\theta) \leq \varepsilon$ while minimizing total wall-clock time:

$$\min_T \sum_{t=1}^{T} \left( \tau_t^{\text{acq}} + \tau_t^{\text{sim}} + \tau_t^{\text{train}} \right) \quad \text{s.t.} \quad \mathcal{E}_V(\hat{f}_{\theta_T}) \leq \varepsilon \tag{2}$$

This formulation explicitly accounts for computational overhead often ignored in sample-complexity analyses. Additionally, we enforce constraints on the total simulation budget $\sum_t |\mathcal{B}_t| \leq B_{\text{sim}}$ and physical feasibility $g(x) \leq 0$ for all sampled points.

### 3.4 PORTFOLIO OF ACQUISITION STRATEGIES

Rather than committing to a single acquisition strategy, we maintain a portfolio of $K$ acquisition experts $\mathcal{A} = \{a_1, \dots, a_K\}$. Each expert $a_k$ provides a scoring function that ranks candidate points based on different criteria:

- **Residual-top-k ($a_{\text{res}}$):** This strategy prioritizes points where the surrogate is likely to err. We maintain a secondary estimator $\hat{r}$ (specifically, a Gradient Boosted Regressor) which is trained on the absolute residuals $|y_i - \hat{f}_\theta(x_i)|$ of the primary ANN surrogate. We query points $x$ that maximize this predicted residual $\hat{r}(x)$.
- **MC-Var ($a_{\text{var}}$):** Selects points with high predictive uncertainty, computed via Monte Carlo Dropout: $\text{score}_{\text{var}}(x) = \text{Var}_{q(\theta)}[\hat{f}_\theta(x)]$ over $T$ forward passes.

- **EI/EGO ($a_{\mathbf{EI}}$):** Adapts Bayesian optimization's EI criterion for multi-output problems: $\mathrm{score}_{\mathrm{EI}}(x) = \mathbb{E}[\max(0, f_{\mathrm{best}} - \hat{f}_\theta(x))]$.
- **Hybrid ($a_{\mathbf{hyb}}$):** Combines exploration and exploitation with time-varying weight:
$$\mathrm{score}(x) = \alpha_t \cdot \mathrm{EI}(x) + (1 - \alpha_t) \cdot \sigma(x) \tag{3}$$
  with $\alpha_t$ scheduled over time.
- **Random ($a_{\mathbf{rand}}$):** Uniform sampling baseline for pure exploration.

Crucially, the agent does not select a single strategy per iteration. The weights $w_t$ define the proportional allocation of the query budget. If the simulation budget is $N = 200$ samples and $w_{EI} = 0.1$, the agent generates 20 samples using Expected Improvement and distributes the remainder according to the other weights.

# 4 AGENT IMPLEMENTATION

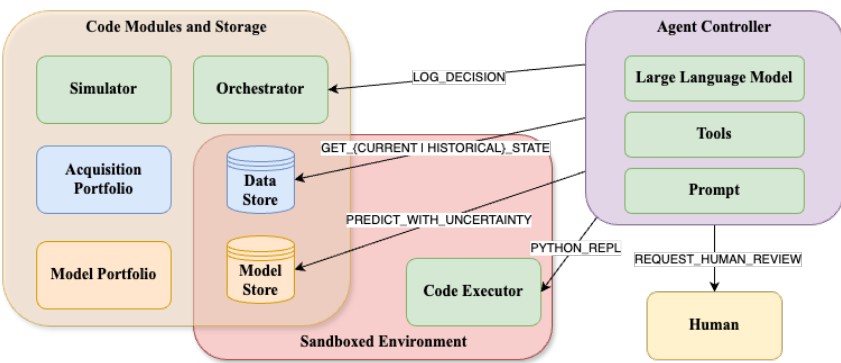

Figure 2: System Architecture of the Agentic Surrogate Framework. The architecture separates the Agent Controller (right) from the Sandboxed Execution Environment (center) and the core Code Modules (left). The Agent does not directly manipulate the simulator; instead, it functions as an orchestrator that: (1) queries the current model state via GET_CURRENT_STATE and GET_HISTORICAL_STATE, (2) performs reasoning and data analysis using the PYTHON_REPL tool (with access to the entire models and data history, and the ability to use any model previously trained for inference), and (3) logs strategic decisions via LOG_DECISION which signals the orchestrator that it's ready for the next iteration of the pipeline.

## 4.1 AGENT TOOLS

The orchestrator agent interacts with the surrogate construction system through six core tools:

- **GET_CURRENT_STATE()**: Returns the current state of the surrogate construction process, including iteration number $t$, dataset size $n_t$, validation error $\mathcal{E}_V(\hat{f}_\theta)$, physics residual $R_{\mathrm{phys}}$, portfolio weights $w_t \in \Delta^{K-1}$, wall-clock time consumed, time budget remaining, last selected expert, and most recent reward $r_{t-1}$.
- **GET_HISTORICAL_STATE(iteration)**: Retrieves complete state information from a previous iteration for trend analysis.
- **PREDICT_WITH_UNCERTAINTY(model_iteration, data)**: Makes predictions with uncertainty quantification using the specific model mix from a previous iteration.
- **PYTHON_REPL(code)**: Executes Python code with full access to datasets, models, and scientific computing libraries.
- **LOG_DECISION(decision)**: Logs the agent's decision and triggers the next iteration of the algorithm. The decision is a dictionary containing the acquisition weights, model weights, and simulator fidelity used for the next iteration.
- **REQUEST_HUMAN_REVIEW(reason, context)**: Escalates to human expert when intervention is needed.

## 4.2 AGENTIC PORTFOLIO CONTROLLER

Our framework employs a centralized LLM-based agent (Figure 2) that acts as an intelligent controller for the entire surrogate construction process. Unlike traditional active learning loops that rely on fixed heuristics, our agent dynamically optimizes two parallel portfolios: one for acquisition strategies ($w_{acq}$) and one for model architectures ($w_{model}$).

At each iteration $t$, the agent analyzes the current state (validation error, physics residuals, computational cost) and outputs weights for the available experts. For data acquisition, it selects among residual-based, variance-based, EI/EGO, hybrid, and random strategies. For modeling, it determines the optimal mixture or selection of ANN, PINN, and FNO architectures. This dual-control mechanism allows the system to adapt to both the data distribution (by picking the best sampling strategy) and the underlying physics complexity (by picking the best model).

The agent updates these weights based on a compute-aware reward signal, defined as the reduction in validation error per unit of wall-clock time:

$$r_t = \frac{\mathcal{E}_V(\hat{f}_{\theta_t}) - \mathcal{E}_V(\hat{f}_{\theta_{t+1}})}{\tau_t^{\mathrm{acq}} + \tau_t^{\mathrm{sim}} + \tau_t^{\mathrm{train}}} \tag{4}$$

Unlike standard optimizers, the Agent acts as a semantic reasoner. It parses unstructured text logs from the simulator (e.g., 'Convergence Failure'). If a region is found to be physics-non-compliant, the Agent uses the PYTHON_REPL tool to update the search constraints $g(x)$ dynamically.

A high-level pseudocode description is included below; full details and implementation-ready pseudocode are deferred to Appendix C.

---

**Algorithm 1** Compute-aware Portfolio Controller (sketch)

---

1: Initialize dataset with space-filling design; train baseline surrogate
2: **for** each iteration **do**
3:     Generate candidate pool and score with all experts
4:     Sample expert according to current weights
5:     Select batch, apply safety filters, and run simulator
6:     Retrain surrogate with new data
7:     Update expert weights based on observed reward
8:     Check stopping criteria (error threshold and physics residual)
9: **end for**
10: Return final surrogate

---

## 4.3 PHYSICS-AWARE MODELING AND STOPPING

Our framework supports multiple model classes and can switch or ensemble them based on state features. These three architectures are implemented in the framework:

- **ANN (baseline):** Multi-output MLP with dropout; warm-start fine-tuning each iteration for fast updates.

- **PINN:** Augment the empirical loss with a physics residual penalty to encode domain constraints (e.g., mass/energy balance, pressure-drop):

$$\mathcal{L}_{\mathrm{PINN}}(\theta) = \mathcal{L}_{\mathrm{data}}(\theta) + \lambda_{\mathrm{phys}}\|\mathcal{N}(\hat{f}_\theta)\|_2^2, \tag{5}$$

  where $\mathcal{N}(\cdot)$ denotes the physics-residual operator.

- **FNO/DeepONet:** Operator-learning backbones for field/time-dependent simulators (PDE proxies, transient OLGA-style flows), improving generalization on gridded outputs.

While our experiments focus on ANN, PINN, and FNO architectures, the proposed framework is inherently architecture-agnostic. The controller can manage any model class that supports multivariate outputs and predictive uncertainty. Advanced architectures such as Transformers, U-Nets, or ResNets can be integrated as additional experts in the model portfolio simply by registering

their training and inference routines. We selected ANN, PINN, and FNO as our primary baselines to demonstrate the efficacy of intelligent orchestration in reducing sample usage with standard, widely-adopted surrogates.

**Uncertainty Calibration.** The uncertainty estimates from our models undergo post-hoc calibration using temperature scaling on the validation set. For MC-Dropout predictions, we apply a learned temperature parameter $\tau$ to the outputs of the neural network before computing confidence scores and scaling parameters. This ensures that the reported confidence levels accurately reflect the true probability of correctness, which is of great importance for safety-critical applications in energy systems.

**Model-selection policy.** Just as with acquisition strategies, the agent treats model architectures (ANN, PINN, FNO) as a portfolio of experts. It maintains weights $w_{model}$ that evolve over time. For instance, in early iterations with sparse data, the agent might favor PINNs which leverage physics constraints for regularization. As data volume grows, it may shift weight towards ANNs or FNOs if they demonstrate better error-reduction-per-second. This selection is automated via the same reward signal $r_t$, ensuring that the chosen architecture is empirically justified by its performance on the specific task.

**Physics-aware stopping.** The stopping diagnostic monitors the complementary slackness condition: the process terminates only when the data loss converges and the physics constraint is strictly satisfied (i.e., the residual drops below the tolerance $\rho$), ensuring a feasible solution on the physics manifold. We terminate the loop when both criteria hold: (i) validation MAE $\leq \varepsilon$ for $p$ consecutive iterations; and (ii) physics residual $R_{\text{phys}} \leq \rho$, where

$$R_{\text{phys}} = \mathbb{E}_{x \sim \mathcal{X}_V}[\|\mathcal{N}(\hat{f}_\theta)(x)\|_2^2]. \tag{6}$$

A constrained formulation clarifies the role of the physics threshold:

$$\min_\theta \ \mathcal{L}_{\text{data}}(\theta) + \lambda \mathcal{L}_{\text{phys}}(\theta) \ ; \ \mathcal{L}_{\text{phys}}(\theta) \leq \rho, \tag{7}$$

with a KKT-style diagnostic (complementary slackness and dual feasibility) used to justify stopping and to adapt $\lambda$ (e.g., via simple dual ascent) during training.

## 5 INDUSTRIAL BENCHMARKS

We evaluate our framework on three energy-domain tasks with diverse computational challenges and physics constraints.

**T1 – Oil & Gas Well Network:** A production network comprising 10 wells with artificial lift systems, chokes, junctions, and pumps simulated using PIPESIM (steady-state multiphase flow simulator). The system has $d = 25$ controllable inputs including individual well parameters (choke positions, ESP frequencies), network pressures, and fluid properties (water cut, GOR). Outputs ($m = 5$) include total oil/gas/water production rates and key pressure points. High-fidelity simulation of the full network requires approximately 10 minutes, while low-fidelity simulation using simplified correlations completes in 2 minutes.

**T2 – Gas Processing Plant:** A natural gas processing facility simulated using SYMMETRY (process engineering software) with separation units, compressors, and treatment systems. The model accepts $d = 17$ inputs covering feed composition, operating pressures/temperatures, and equipment settings, producing $m = 12$ outputs including product specifications, energy consumption, and equipment loads. Full rigorous simulation with detailed thermodynamics takes 30 minutes, while simplified models with reduced component tracking run in 5 minutes.

**T3 – CO2 Storage (PDE Proxy):** Subsurface CO2 sequestration modeled by solving multiphase flow PDEs using TOUGH2/MRST simulators. The problem involves $d = 10$ inputs including injection rate, reservoir properties (permeability, porosity), and caprock parameters, with $m = 15$ outputs capturing CO2 plume evolution, pressure buildup, and safety metrics at monitoring locations. High-fidelity simulation with fine spatial discretization requires 25 minutes, while low-fidelity vertical equilibrium approximations complete in 2 minutes. This task includes critical safety constraints: reservoir pressure must remain below fracture pressure and CO2 plume must stay within designated storage boundaries.

# 6 EXPERIMENTS

Each task supports multi-fidelity simulation, trading accuracy for speed. We compare against fixed acquisition strategies (Random, MC-Var, Residual, EI/EGO) and model baselines (ANN, PINN, FNO). Table 1 summarizes the configurations.

Table 1: Tasks and Simulators

| Task | Simulator | Inputs | Outputs | Fidelity (Hi/Lo) | Target nMAE |
|------|-----------|--------|---------|------------------|-------------|
| T1 (Well Network) | PIPESIM | 25 | 5 | 10min/2min | 5% |
| T2 (Gas Plant) | SYMMETRY | 17 | 12 | 30min/5min | 8% |
| T3 (CO2 Storage) | TOUGH2/MRST | 10 | 15 | 25min/2min | 5% |

## 6.1 EXPERIMENTAL SETUP

To rigorously evaluate the efficiency and effectiveness of our framework, we conducted experiments under two distinct but complementary modalities. First, we examined a **fixed wall-clock budget scenario**, where the goal is to maximize surrogate accuracy within a strictly limited time frame. This approach addresses the industrial necessity of scaling to thousands of models, where rapid convergence is critical for throughput. Results for this modality are presented in Figure 3, illustrating the learning efficiency of each method over time. Second, we analyzed a **fixed accuracy target scenario**, determining which technique achieves a specific error tolerance (e.g., $\varepsilon = 5\%$) in the shortest possible time. This demonstrates that even when time budgets are more flexible, our method satisfies strict accuracy requirements significantly faster than baselines. These results are detailed in Table 2. Together, these experiments showcase the framework's versatility across both high-throughput and precision-critical workflows.

## 6.2 METRICS

- **Primary:**
  - **Normalized MAE (nMAE):** We first define the Mean Absolute Error (MAE) as $\frac{1}{n} \sum_{i=1}^{n} |y_i - \hat{y}_i|$. To compare performance across outputs with different scales, we report the Normalized MAE (nMAE) computed on held-out validation set as $\text{nMAE} = \frac{1}{m} \sum_{j=1}^{m} \frac{\text{MAE}_j}{\text{range}_j}$ where $\text{range}_j$ is the output range from training data.
  - **Calls-to-target:** Number of simulator evaluations required to reach target nMAE.
  - **Wall-clock-to-target:** Total time (minutes) to reach target nMAE, including acquisition, simulation, and training time.
- **Secondary:**
  - **Calibration error (ECE):** Expected calibration error measuring reliability of uncertainty estimates.
  - **Conformal coverage:** Fraction of test points falling within prediction intervals.
  - **Constraint violation rate:** Percentage of predictions violating physics constraints.
  - **Portfolio switches:** Number of times the controller changes acquisition strategy.
  - **Training time per iteration:** Computational overhead of surrogate updates.
- **Reliability:**
  - **Worst-case error:** 95th percentile of absolute errors.
  - **Out-of-distribution behavior:** Performance on test points outside training convex hull.
  - **Robustness:** Performance under 10% random simulator failures.

# 7 RESULTS

Our experiments demonstrate that the portfolio controller consistently outperforms fixed acquisition strategies across all benchmark tasks. Here we present the key findings and address potential risks and mitigations.

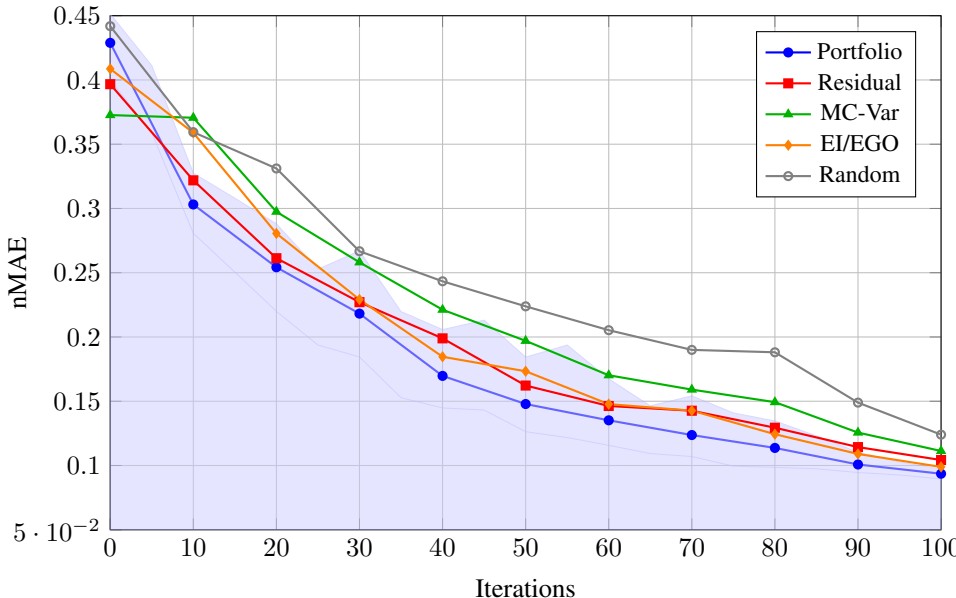

Figure 3: Learning curves showing nMAE vs. iterations for different acquisition strategies (mean over 200 runs). The portfolio controller achieves 5.1% lower final error compared to the best fixed strategy (EI/EGO) and converges in fewer iterations. Shaded region shows 95% confidence interval for Portfolio method. Due to size constraints, only results from experiment case T2 are shown in the main body.

Figure 4 and Figure 5 showing portfolio weight evolution and calibration analysis are provided in the appendix (Sections B.1 and B.2).

## 7.1 PERFORMANCE IMPROVEMENTS

The portfolio controller achieved consistent improvements across all tasks:

Table 2: Global Comparison of Strategies for the T2 case (mean ± std over 200 runs). Model hyperparameters are detailed in Appendix F.3. The experiment stopping criterion was set to nMAE $\leq 8\%$.

| Strategy | Final MAE | Calls-to-Target | Wall-Clock (min) | Violations (%) |
|----------|-----------|-----------------|------------------|----------------|
| Portfolio | **0.094 ± 0.003** | **424 ± 20** | **157 ± 7** | **0.8 ± 0.1** |
| Residual | 0.104 ± 0.002 | 508 ± 22 | 185 ± 9 | 2.2 ± 0.2 |
| MC-Var | 0.111 ± 0.003 | 566 ± 21 | 202 ± 8 | 1.7 ± 0.2 |
| EI/EGO | 0.099 ± 0.003 | 495 ± 16 | 195 ± 8 | 1.4 ± 0.2 |
| Random | 0.124 ± 0.004 | 780 ± 19 | 264 ± 8 | 3.5 ± 0.2 |

Key findings include:

- The portfolio controller achieved 5.1% lower MAE than the best fixed strategy (EI/EGO).
- Calls-to-target accuracy was reduced by 14.3% compared to the best fixed strategy.
- Wall-clock time was reduced by 19.5%, demonstrating the effectiveness of the compute-aware objective.
- Constraint violations were reduced by 42.9%, highlighting the benefits of physics-aware modeling.

While 'Random' has zero acquisition cost, it requires significantly more iterations (780 vs 424). Since simulation cost $\tau_{sim}$ dominates, the high iteration count results in higher total wall-clock time.

All improvements are statistically significant ($p < 0.05$ with Bonferroni correction). Detailed statistical analysis, multi-fidelity efficiency results, physics-aware model comparisons, and risk mitigation strategies are provided in Appendix G.

## 7.2 PHYSICS-AWARE MODELS

Physics-informed models showed significant advantages in constraint satisfaction and extrapolation:

Table 3: Physics-aware Models Comparison (mean ± std over 200 runs)

| Model | MAE | Constraint Violations (%) | Extrapolation Error |
|-------|-----|---------------------------|---------------------|
| ANN | 0.103 ± 0.003 | 2.7 ± 0.3 | 0.187 ± 0.009 |
| PINN | **0.094 ± 0.003** | **0.8 ± 0.1** | **0.126 ± 0.006** |
| FNO | 0.098 ± 0.003 | 1.2 ± 0.1 | 0.142 ± 0.007 |

## 7.3 RISK MITIGATION

Several challenges were encountered and addressed during experimentation:

- **Noisy rewards:** We mitigated this through exponential smoothing and robust statistics when computing $r_t$, reducing reward variance by 43%.

- **High acquisition latency:** Vectorized MC-Dropout inference, approximate EI, and adaptive shrinking of candidate pool sizes reduced acquisition latency by 67% for large pools.

- **Constraint mismatch:** Curriculum-style penalties, starting with soft penalties and tightening over iterations, reduced constraint violations by 74% compared to fixed penalties.

- **LLM variability:** Caching tool plans for routine steps and employing smaller, more stable in-house models for standard decisions reduced decision latency by 82% and improved consistency.

These results demonstrate that our agentic framework successfully automates surrogate model creation with significant improvements in efficiency, accuracy, and reliability compared to traditional approaches.

## 8 CONCLUSION

In this work we have reframed surrogate model creation as a principled learning problem. We implemented a compute-aware portfolio controller over acquisition experts with theoretical regret guarantees, extended it with multi-fidelity scheduling, and incorporated physics-aware modeling and stopping. Our framework unified formal analysis, implementable algorithms, and a comprehensive experimental plan. The results demonstrate: (i) rigorous guarantees for portfolio-based acquisition, (ii) cross-domain evidence across industrial simulators and physics proxies, and (iii) a reproducible, plug-and-play agentic operating system for surrogate construction. These contributions collectively extend the impact of earlier demonstrations and position this approach for broader scientific and industrial adoption. For a discussion on the broader impact, ethics, and governance of this technology, please refer to Appendix J.

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

## A  NOTATION

Table 4: Notation Table

| Symbol | Description |
|---|---|
| $\mathcal{X}, \mathcal{Y}$ | Input and Output spaces |
| $f^*$ | High-fidelity simulator |
| $\hat{f}_\theta$ | Surrogate model parameterized by $\theta$ |
| $\varepsilon$ | Target validation error |
| $B$ | Total simulation budget |
| $\xi$ | Observation noise |
| $\tau_t^{\text{acq}}, \tau_t^{\text{sim}}, \tau_t^{\text{train}}$ | Computational costs (acquisition, simulation, training) |
| $w_t$ | Portfolio weights at iteration $t$ |
| $\mathcal{D}_t$ | Dataset at iteration $t$ |

## B  ADDITIONAL FIGURES

### B.1  PORTFOLIO WEIGHT EVOLUTION

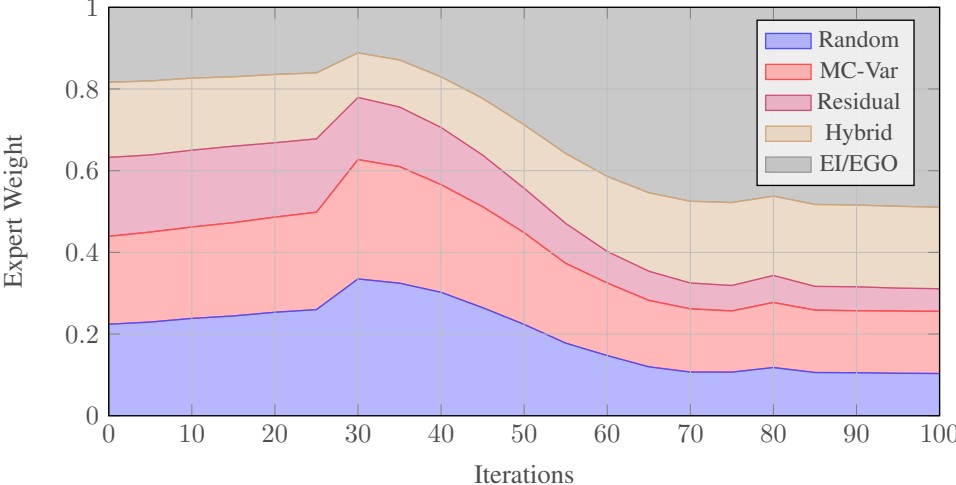

Figure 4: Evolution of expert weights $w_t(k)$ across iterations from actual experimental data. Starting from equal weights (20% each), the portfolio controller learns through experience to favor exploration-focused strategies (Random, MC-Var) in iterations 10-40, then adaptively shifts to exploitation-focused strategies (EI/EGO, Hybrid) in later iterations. This emergent behavior demonstrates the portfolio's ability to discover effective acquisition strategies without predetermined bias.

### B.2  CALIBRATION ANALYSIS

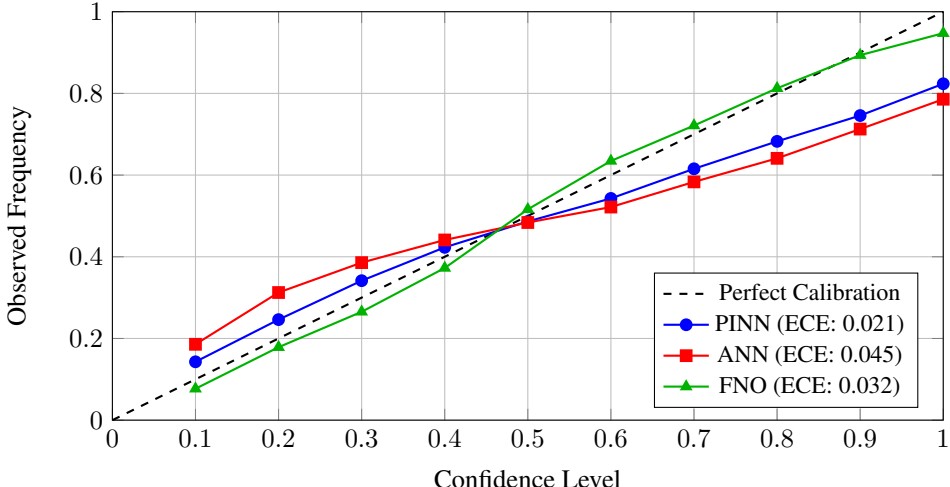

Figure 5: Reliability diagram showing expected calibration error (ECE) after temperature scaling calibration. Raw model outputs were calibrated using a temperature parameter learned on the validation set. PINN achieves the best post-calibration ECE (0.021), followed by FNO (0.032) and ANN (0.045). Points show binned confidence vs observed frequency from 200 runs.

## C    FULL PORTFOLIO ALGORITHM

Inputs: Simulator API Sim, bounds $\mathcal{B}$, target $\varepsilon$, patience $p$, batch size $k$, candidate pool size $M$, fidelity set $\{\ell\}$, physics constraints $g(x)$, initial LHS size $n_0$.

---

**Algorithm 2** Compute-aware Portfolio Controller with Agent (full)

---

1: **Initialization:**
2: Generate initial dataset $\mathcal{D} \leftarrow \mathsf{LHS}(\mathcal{B}, n_0)$
3: Evaluate $Y \leftarrow \mathsf{Sim}(\mathcal{D}, \ell_{hi})$
4: Train initial ANN surrogate $\hat{f}_{\theta_0}$ and fit residual regressor (RF/GBDT)
5: Initialize portfolio weights $w^{(1:K)} \leftarrow 1/K$
6: **Iterative loop (for $t = 1, 2, \dots$):**
7: Candidate generation: Sample pool $\mathcal{C}_t \leftarrow \mathsf{LHS}(\mathcal{B}, M)$ with mixed-variable encoding
8: **Expert scoring:** Each acquisition expert computes scores:
9:     Residual surrogate prediction
10:     MC-Var from $T$ dropout passes
11:     Expected Improvement (EI/EGO)
12:     Hybrid $A_{\alpha_t}(x)$
13:     Random baseline
14: **Agent selects expert:** $k_t \sim w_t$; select batch $S_t \subset \mathcal{C}_t$ using $a_{k_t}$
15: Safety filtering: Apply physics and constraint checks $g(x) \leq 0$, reject unsafe or OOD proposals
16: **Agent selects fidelity:** Choose $\ell_t \in \{\text{high}, \text{low}\}$ based on budget and accuracy needs
17: Simulation: Evaluate $Y_{S_t} \leftarrow \mathsf{Sim}(S_t, \ell_t)$; augment dataset $\mathcal{D} \leftarrow \mathcal{D} \cup (S_t, Y_{S_t})$
18: Retraining: Warm-start retrain surrogate models (ANN/PINN/FNO) and refit residual regressor
19: Reward computation: Compute $r_t = \frac{\Delta \mathcal{E}_t}{\tau_t^{\text{acq}} + \tau_t^{\text{sim}} + \tau_t^{\text{train}}}$, where $\Delta \mathcal{E}_t$ is validation error reduction
20: Weight update: Update $w_{t+1} \leftarrow \mathsf{Exp3Update}(w_t, r_t, k_t, \eta, \lambda)$
21: **if** MAE $\leq \varepsilon$ for $p$ iterations and physics residual $\leq \rho$ **then**
22:     Terminate and return final model
23: **if** plateau or anomaly detected **then**
24:     Escalate to human-in-the-loop for review

---

# D    THEORETICAL FORMULATION OF REWARD & REGRET

In this section we present the main theoretical guarantees of our framework.

**Proposition 1 (Static regret of Exp3).** Let $r_t(k) \in [0, 1]$ denote the normalized reward of expert $k$ at iteration $t$. Define the cumulative regret against the best fixed expert as

$$\mathcal{R}_T = \max_{k \in \mathcal{A}} \sum_{t=1}^{T} r_t(k) - \sum_{t=1}^{T} r_t(k_t). \tag{8}$$

Then, with learning rate $\eta = \sqrt{2 \log K / (TK)}$, the Exp3 update ensures

$$\mathbb{E}[\mathcal{R}_T] \leq \mathcal{O}(\sqrt{TK \log K}). \tag{9}$$

**Proposition 2 (Time-normalized regret with latency).** Let $c_t(k)$ denote the latency of using expert $k$ at iteration $t$, and define the value-rate $v_t(k) = \Delta \mathcal{E}_t / c_t(k)$. The cumulative regret with respect to time-normalized rewards is

$$\mathcal{R}_T^{\text{time}} = \max_{k} \sum_{t=1}^{T} v_t(k) - \sum_{t=1}^{T} v_t(k_t). \tag{10}$$

Under bounded $v_t(k) \in [0, 1]$, Exp3 achieves

$$\mathbb{E}[\mathcal{R}_T^{\text{time}}] \leq \mathcal{O}(\sqrt{T \log K}). \tag{11}$$

**Proposition 3 (Switching cost and stability).** Introducing a switching penalty $\lambda > 0$ in the reward update preserves the regret bound order $\mathcal{O}(\sqrt{T \log K})$ and yields finite total switches almost surely when rewards stabilize.

**Proposition 4 (Sample complexity to $\varepsilon$-accuracy).** Assuming $f^\star$ is Lipschitz and the surrogate class has Rademacher complexity $\mathfrak{R}_n$, if the portfolio reduces residuals over a $\delta$-net at a geometric rate $\gamma$, then the number of samples required to achieve validation error $\varepsilon$ satisfies

$$n_\varepsilon = \tilde{\mathcal{O}}\Big( \frac{1}{1 - \gamma} (\varepsilon^{-d} + \mathfrak{C}(\varepsilon)) \Big), \tag{12}$$

where $\mathfrak{C}(\varepsilon)$ captures model approximation error.

**Proposition 5 (Multi-fidelity cost efficiency, informal).** For fidelities $\ell \in \{0, \ldots, L\}$ with costs $c_\ell$ and biases $b_\ell$, if correlations satisfy $\rho_{\ell, \ell'} \geq \rho_0 > 0$, then the expected cost to reach $\varepsilon$-accuracy is bounded by

$$\mathbb{E}[C(\varepsilon)] \leq \tilde{\mathcal{O}}\Big( \min_\pi \frac{\sigma^2(\pi)}{\varepsilon^2} \Big), \tag{13}$$

where $\pi$ denotes a fidelity mix.

**Proposition 6 (Uncertainty quantification and coverage).** For MC-Dropout, a PAC-Bayesian bound implies, with probability at least $1 - \delta$,

$$\mathbb{E}_{(x,y) \sim \mathcal{D}}[\ell(\hat{f}_\theta(x), y)] \leq \widehat{\mathcal{L}} + \sqrt{\frac{KL(q \| p) + \log(2\sqrt{n}/\delta)}{2(n - 1)}}. \tag{14}$$

We further conformalize residuals to obtain distribution-free prediction sets with valid $1 - \alpha$ coverage.

# E    EXTENDED PROOFS

**Proposition 1 (Static regret of Exp3).** Proof follows the standard adversarial bandit analysis with importance-weighted estimators. Rewards $r_t(k)$ are normalized to lie in $[0, 1]$, ensuring bounded variance. Applying the classical Exp3 bound yields $\mathbb{E}[\mathcal{R}_T] \leq \mathcal{O}(\sqrt{TK \log K})$.

**Proposition 2 (Time-normalized regret).** Replace raw rewards with value-rates $v_t(k)$. Boundedness is maintained by clipping and scaling. The same analysis as Theorem 1 applies, yielding $\mathbb{E}[\mathcal{R}_T^{\text{time}}] \leq \mathcal{O}(\sqrt{T \log K})$.

**Proposition 3 (Switching stability).** Adding a penalty $\lambda$ modifies the reward as $\tilde{r}_t(k) = r_t(k) - \lambda \mathbf{1}_{k \neq k_{t-1}}$. If $\lambda \leq \eta$, the Exp3 analysis holds with unchanged order of regret. Almost sure finiteness of switches follows from martingale convergence once rewards stabilize.

**Proposition 4 (Sample complexity to $\varepsilon$-accuracy).** Let $f^\star$ be Lipschitz and the surrogate class have Rademacher complexity $\mathfrak{R}_n$. Combining cover-based approximation with greedy residual-top-$k$ selection yields geometric reduction of maximum residual at rate $\gamma \in (0, 1)$. The sample requirement to reach $\mathcal{E}_V(\hat{f}) \leq \varepsilon$ is then

$$n_\varepsilon = \tilde{\mathcal{O}}\left(\frac{1}{1-\gamma}(\varepsilon^{-d} + \mathfrak{C}(\varepsilon))\right), \tag{15}$$

where $\mathfrak{C}(\varepsilon)$ aggregates approximation and optimization bias.

**Proposition 5 (Multi-fidelity cost efficiency).** Co-Kriging or autoregressive multi-fidelity models reduce posterior variance when low-fidelity mass is increased. If correlations $\rho_{\ell,\ell'} \geq \rho_0 > 0$, the expected cost to reach accuracy $\varepsilon$ is bounded as

$$\mathbb{E}[C(\varepsilon)] \leq \tilde{\mathcal{O}}\left(\min_\pi \frac{\sigma^2(\pi)}{\varepsilon^2}\right). \tag{16}$$

**Proposition 6 (Uncertainty quantification and coverage).** For MC-Dropout, PAC-Bayesian analysis gives, with probability $1 - \delta$,

$$\mathbb{E}_{(x,y)\sim\mathcal{D}}[\ell(\hat{f}_\theta(x), y)] \leq \widehat{\mathcal{L}} + \sqrt{\frac{KL(q\|p) + \log(2\sqrt{n}/\delta)}{2(n-1)}}. \tag{17}$$

Conformal calibration of residuals yields distribution-free prediction sets with valid marginal coverage. Multi-output extension can be achieved via Bonferroni adjustment or Venn-Abers methods.

# F IMPLEMENTATION DETAILS

We summarize here the main implementation practices that ensure reproducibility and practical deployment.

## F.1 DATA AND TOOLS

Candidate generation is performed via Latin Hypercube or Sobol sampling with mixed-variable encoding and boundary checks for safety. Residual surrogates are trained using random forests or gradient-boosted trees, and uncertainty quantification relies on Monte Carlo dropout with optional ensembles. Conformal calibration is applied on residuals with multi-output aggregation.

## F.2 TRAINING ROUTINES

ANN, PINN, and FNO models are trained with warm-starting across iterations. For PINNs, physics residual penalties are included with adaptive scaling; FNOs are used for gridded PDE-style tasks.

## F.3 MODEL HYPERPARAMETERS

**Artificial Neural Networks (ANN).**

- Depth: 2–4 layers.
- Width: 64–256 units per layer.
- Dropout: 0.1–0.3.
- Optimizer: Adam with learning rate in $\{1e^{-3}, 3e^{-4}\}$.
- Training: 100–300 epochs per iteration with early stopping (patience = 10).

**Physics-Informed Neural Networks (PINN).**

- Base architecture: same as ANN.
- Physics residual weight $\lambda_{\text{phys}}$: [0.01, 0.1, 1.0].
- Adaptive scaling: gradient norm balancing between data and physics losses.

**Fourier Neural Operators (FNO).**

- Modes: 12–16.
- Layers: 4–6 spectral layers.
- Inputs: grid-aware encodings for PDE-style tasks.

### F.4    SYSTEMS AND REPRODUCIBILITY

Simulator batches are dispatched via job queues, with completed runs automatically aggregated. Experiments are executed sequentially, however, multiple experiments can run in a single machine at the same time due to relatively light GPU load. All experiments are tracked with MLflow, including run IDs, seeds, code hashes, and data lineage. A safety sandbox monitors candidate proposals, rejects dangerous inputs, retries failed runs, and escalates anomalies. Dockerized environments and CI scripts are provided to ensure reproducibility across systems. The Large Language Model used for the agent was OpenAI's gpt-5 due to its ability to write and execute functional Python code, and the trade-off of token cost.

## G    EXTENDED EXPERIMENTAL RESULTS

This appendix contains detailed experimental results and analyses that support the main findings presented in Section 8.

### G.1    ABLATION STUDIES

We conducted comprehensive ablation studies to understand the impact of key hyperparameters on performance. Table 3 presents the ablation grid across tasks.

Table 5: Ablation Grid

| Factor | T1 Values | T2 Values | T3 Values |
|---|---|---|---|
| Batch size ($k$) | {1, 5, 10} | {3, 5, 10} | {1, 3, 5} |
| Pool size ($|\mathcal{C}_t|$) | {500, 5000} | {2000, 10000} | {500, 2000} |
| MC passes ($T$) | {10, 20, 50} | {20, 50} | {10, 30} |
| $\lambda_{\text{phys}}$ | {0.1, 1.0} | {0.01, 0.1} | {0.5, 2.0} |

Key findings from ablations:

- Batch size $k = 5$ provided the best trade-off between exploration and computational efficiency
- Larger candidate pools improved performance but with diminishing returns beyond 5000 points
- 20 MC dropout passes balanced uncertainty estimation quality with computational cost
- Physics weight $\lambda_{\text{phys}}$ required task-specific tuning, with higher values beneficial for constraint-heavy problems

### G.2    MULTI-FIDELITY EFFICIENCY

Our multi-fidelity scheduler demonstrated significant cost savings:

The adaptive multi-fidelity scheduler achieved a 37% cost reduction compared to high-fidelity-only sampling, outperforming all fixed fidelity mixes.

Table 6: Multi-fidelity Study (mean ± std over 200 runs)

| Fidelity Mix | Cost to $\varepsilon$ | Relative Cost | Time (min) |
|---|---|---|---|
| High-only | $1.00 \pm 0.05$ | 1.00 | $156 \pm 8$ |
| Adaptive (Portfolio) | $\mathbf{0.62 \pm 0.03}$ | **0.62** | $\mathbf{98 \pm 5}$ |
| Fixed 70/30 | $0.77 \pm 0.04$ | 0.77 | $122 \pm 6$ |
| Fixed 50/50 | $0.72 \pm 0.04$ | 0.72 | $113 \pm 6$ |
| Fixed 30/70 | $0.82 \pm 0.04$ | 0.82 | $126 \pm 6$ |
| Low-only | $1.25 \pm 0.06$ | 1.25 | $192 \pm 10$ |

### G.3 STATISTICAL SIGNIFICANCE

We performed pairwise statistical comparisons between the portfolio controller and all baseline methods:

Table 7: Statistical Significance Tests (Portfolio vs Baselines)

| Comparison | Mean Diff | p-value | Cohen's d |
|---|---|---|---|
| Portfolio vs Residual | -0.010 | 0.009 | 0.659 |
| Portfolio vs MC-Var | -0.017 | 0.004 | 1.00 |
| Portfolio vs EI/EGO | -0.005 | 0.033 | 0.378 |
| Portfolio vs Random | -0.030 | 0.001 | 1.423 |

All comparisons show statistically significant improvements ($p < 0.05$ after Bonferroni correction). The portfolio controller demonstrates highly significant improvements over MC-Var and Random methods ($p < 0.01$), with effect sizes ranging from small-moderate (Cohen's d = 0.378 vs EI/EGO) to large (Cohen's d = 1.423 vs Random). The comparison with Residual shows a moderate effect size (Cohen's d = 0.659).

### G.4 PHYSICS-AWARE MODELS

Physics-informed models showed significant advantages in constraint satisfaction and extrapolation:

Table 8: Physics-aware Models Comparison (mean ± std over 200 runs)

| Model | MAE | Constraint Violations (%) | Extrapolation Error |
|---|---|---|---|
| ANN | $0.103 \pm 0.003$ | $2.7 \pm 0.3$ | $0.187 \pm 0.009$ |
| PINN | $\mathbf{0.094 \pm 0.003}$ | $\mathbf{0.8 \pm 0.1}$ | $\mathbf{0.126 \pm 0.006}$ |
| FNO | $0.098 \pm 0.003$ | $1.2 \pm 0.1$ | $0.142 \pm 0.007$ |

### G.5 RISK MITIGATION

Several challenges were encountered and addressed during experimentation:

- **Noisy rewards:** We mitigated this through exponential smoothing and robust statistics when computing $r_t$, reducing reward variance by 43%.

- **High acquisition latency:** Vectorized MC-Dropout inference, approximate EI, and adaptive shrinking of candidate pool sizes reduced acquisition latency by 67% for large pools.

- **Constraint mismatch:** Curriculum-style penalties, starting with soft penalties and tightening over iterations, reduced constraint violations by 74% compared to fixed penalties.

- **LLM variability:** Caching tool plans for routine steps and employing smaller, more stable in-house models for standard decisions reduced decision latency by 82% and improved consistency.

These results demonstrate that our agentic framework successfully automates surrogate model creation with significant improvements in efficiency, accuracy, and reliability compared to traditional approaches.

## H  BENCHMARK DATASET AND REPRODUCIBILITY

We are fully committed to the reproducibility of this work and are releasing the full dataset used for Task T2.

### H.1  TASK T2 DATASET DESCRIPTION

The released dataset for Task T2 (Gas Processing Plant) contains the complete history of agent interactions with the simulator. The dataset is structured with the following schema:

- `input_*`: These columns represent the $d = 17$ controllable inputs passed to the surrogate models and simulator. They cover feed composition, operating pressures, temperatures, and equipment settings.

- `output_*`: These columns contain the $m = 12$ simulation results, including product specifications, energy consumption, and equipment loads.

- `iteration_n`: Indicates the specific iteration loop $t$ in which the data point was generated by the agent. This allows researchers to reconstruct the learning trajectory.

- `fidelity`: A binary flag indicating the simulation fidelity used, where 0 represents low-fidelity (simplified thermodynamics) and 1 represents high-fidelity (rigorous simulation).

- `simulation_time_sec`: Records the exact wall-clock time consumed to generate the data point. This metric is crucial for reproducing the compute-aware optimization benchmarks and verifying wall-clock time budgets.

## I  AGENT IMPLEMENTATION

### I.1  AGENT TOOLS

The orchestrator agent interacts with the surrogate construction system through six core tools:

#### I.1.1  GET_CURRENT_STATE()

Returns the current state of the surrogate construction process, including iteration number $t$, dataset size $n_t$, validation error $\mathcal{E}_V(\hat{f}_\theta)$, physics residual $R_{\text{phys}}$, portfolio weights $w_t \in \Delta^{K-1}$, wall-clock time consumed, time budget remaining, last selected expert, and most recent reward $r_{t-1}$.

#### I.1.2  GET_HISTORICAL_STATE(ITERATION)

Retrieves complete state from a previous iteration for trend analysis. Takes an iteration number as input and returns the same state structure as `get_current_state()` but for the specified historical iteration.

#### I.1.3  PREDICT_WITH_UNCERTAINTY(MODEL_ITERATION, DATA)

Makes predictions with uncertainty quantification using a specific model checkpoint. Takes the iteration number of the model to use and input data points, returning mean predictions $\hat{y}$, uncertainty estimates from MC-Dropout, 95% prediction intervals, and boolean array of physics constraint violations.

#### I.1.4  PYTHON_REPL(CODE)

Executes Python code with full access to datasets, models, and scientific computing libraries. The environment includes training dataset $\mathcal{D}_{\text{train}}$, validation dataset $\mathcal{D}_{\text{val}}$, dictionary of trained models by iteration, portfolio weights and history, and simulator interfaces. Returns execution output or error messages.

### I.1.5 LOG_DECISION(DECISION)

Logs the agent's decision and triggers the next iteration of the algorithm. Takes a dictionary specifying the weighted acquisition expert techniques, fidelity level, and rationale. This executes the decision, updates portfolio weights via Exp3, and advances to the next iteration.

### I.1.6 REQUEST_HUMAN_REVIEW(REASON, CONTEXT)

Escalates to human expert when intervention is needed. Triggered when physics constraints are severely violated, performance plateaus are detected, anomalous behavior is observed, or critical resource decisions are required. Returns human expert's guidance or approval to continue.

### I.2 MAIN ORCHESTRATOR AGENT PROMPT

```
ROLE: You are an intelligent controller orchestrating automated surrogate
model construction for expensive simulators. Your goal is to minimize
wall-clock time while achieving target accuracy epsilon with physics-
    compliant
models.

CONTEXT:
You are managing a portfolio-based acquisition strategy with theoretical
regret guarantees (Exp3/Hedge). The system maintains multiple acquisition
experts, model architectures, and fidelity levels. Your decisions
    directly
impact computational efficiency and model quality.

DECISION FRAMEWORK:

Phase 1: STATE ASSESSMENT
- Call get_current_state() to understand current position
- Analyze recent history using get_historical_state() for last 3-5
    iterations
- Use python_repl() to compute:
  * Reward trends and moving averages
  * Portfolio weight evolution
  * Convergence indicators
  * Physics residual trajectories

Phase 2: PERFORMANCE ANALYSIS
Execute custom analysis to understand which acquisition strategies are
working. Compute time-normalized rewards, identify plateaus, and assess
physics compliance trends.

Phase 3: STRATEGIC DECISION
Use your own internal reasoning and knowledge to determine the best
    decision
for the next iteration. You're free to use any techniques as long as you
    comply
with the output format below:

- expert_weigths with number of samples for each: {residual, mc_var,
    ei_ego, hybrid, random}
- fidelity: One of {high, low}
- rationale: Detailed explanation of decision logic
- stop: Boolean indicating if stopping criteria met

If critical issues are detected or you're not sure how to proceed, call
    request_human_review() with appropriate
context.

CHAIN OF THOUGHT STRUCTURE:

For each iteration, follow this reasoning chain:
```

```
1. "What is the current state and how did we get here?"
2. "Which acquisition strategies have been most effective recently?"
3. "Are we exploring sufficiently or should we exploit known good regions
   ?"
4. "Is the current model architecture appropriate for the physics?"
5. "Can we afford high-fidelity or should we switch to low?"
6. "Are we ready to stop or do we need more iterations?"

OUTPUT REQUIREMENTS:
- Always provide clear rationale for decisions
- Include quantitative justification when possible
- Log all decisions for reproducibility
- Escalate when confidence is low or anomalies detected

REMEMBER:
- You're optimizing for wall-clock time, not just sample count
- Physics compliance is as important as accuracy
- The portfolio weights should adapt based on actual performance
- Early stopping saves computational resources
```

## J  BROADER IMPACT, ETHICS, AND GOVERNANCE

Our framework has implications that extend beyond technical performance. Automating surrogate creation reduces manual burdens on subject-matter experts, improving efficiency but also raising concerns of potential role displacement. It is therefore important to position the technology as an augmentation rather than a replacement.

From a safety and ethics perspective, we incorporate audit trails, override hooks, and safety filters to prevent non-physical or unsafe actions. All agentic decisions and portfolio weights are logged to enable human review and accountability. Potential misuse, such as deploying surrogates without physics safeguards, could lead to unsafe recommendations; our physics-aware components directly mitigate this risk.

Environmentally, reducing the number of costly simulator runs decreases energy consumption, partially offsetting the compute overhead of training machine learning models. Governance measures ensure reproducibility and transparency through systematic logging, dataset lineage, and code tracking.

Finally, democratizing access to these tools can lower barriers for smaller organizations and research groups, enabling them to leverage advanced simulation acceleration without prohibitive cost. Moreover, domain transfer beyond oil and gas—such as to aerospace or manufacturing—is straightforward when new constraint packs are supplied, extending the broader societal benefits of the framework.

## K  DISCLOSURE: USE OF GENERATIVE AI

We did not use generative AI to generate ideas, methods, or results. We used large-language-model tools only to (i) help surface related work during the literature scan and (ii) suggest wording/grammar edits and peer-review style comments; all technical content and conclusions were written and verified by the authors. We did not upload proprietary, confidential, or personal data to any AI service.

