# OpenReview forum: "Agentic Surrogates: Automating Proxy Models of Simulators with Compute Aware Intelligence"
_ICLR.cc/2026/Conference — Submitted to ICLR 2026_

### Official Review · Reviewer_7saM · 2025-10-21

**Soundness:** 2
**Presentation:** 1
**Contribution:** 2
**Rating:** 2
**Confidence:** 3

**Summary:**

The paper aims to automate the creation of surrogate (proxy) models for complex physics-based simulations using an active learning framework. The setup assumes access to a high-fidelity simulator that is expensive to run, and the goal is to approximate it efficiently with learned surrogates.

The authors propose a mixture-of-experts framework for active learning, where several surrogate models and acquisition strategies are combined adaptively. The combination is governed by a reward-based weighting mechanism called Hedge-Exp3 rules, which balance the contribution of each expert according to their performance.

They also introduce both a cost model and a reward function to guide the optimization:
- The reward is defined as the improvement in residual error divided by the computational time required.
- The cost function aims to minimize the total time for data acquisition, simulation, and training, subject to constraints on validation error, total simulation budget, and physical feasibility of the resulting surrogate.

The method is applied to three oil and gas engineering problems:
- Oil and gas well network modeling,
- Gas processing plant optimization, and
- CO₂ storage simulation.

Across these applications, the proposed method shows improved accuracy, data efficiency (fewer data points required), and faster convergence compared to baseline approaches.

**Strengths:**

The paper tackles an important problem, automating surrogate modeling, where computational cost is often a bottleneck. Introducing a mixture-of-experts framework in this context is an appealing and conceptually sound idea, as it allows for adaptive combination of different surrogate and acquisition strategies. The use of active learning to guide model refinement is also well-motivated and relevant for this class of applications.

A notable strength lies in the practical relevance of the chosen case studies. The three applications are realistic and industrially significant. This grounding in concrete engineering problems makes the work compellin, helps demonstrate potential real-world value and adds credibility.

**Weaknesses:**

The main weakness of the paper lies in its poor writing and presentation quality, which makes it extremely difficult to understand the actual contributions and methodology. Key concepts are often introduced abruptly, without proper motivation, definitions, or notation. This affects the overall readability and credibility of the work.

1. Lack of Clarity and Structure
    - The paper frequently introduces technical components, such as the cost model, portfolio acquisition strategy, or active learning setup, without explanation or contextualization.
    - Important terms like active learning, Hedge-Exp3, or even core notations (e.g., ε, ρ) are never defined or discussed before being used.
    - There is no clear description of how iterations are performed or what the stopping criterion is.
    - Several sections (e.g., “metrics” section 6.1) are underdeveloped, some consisting of a single sentence, while key elements (like the oil and gas problem definitions) are relegated to the appendix.

2. Poor Methodological Explanation
    - The acquisition strategies are not well explained. For instance, the “residual predictor” and “residual top-k” methods are mentioned without detailing how they operate.
    - It is unclear from the main text how candidate points are generated at each iteration, and this information only appears in the appendix.
    - The role of the reward and cost model is not clearly established in the algorithmic flow.
    - The Hedge-Exp3 weighting mechanism is introduced but never properly explained — the reader is left unsure how the weights are applied (e.g., sampling vs. linear combination).
    - The mixture-of-experts setup itself is opaque: it’s not clear whether surrogate models are combined, selected, or average. Same for acquisition experts.
    - The function $g$ similarly is never explained

3. Theoretical Weaknesses
    - The theoretical section lacks substance. The proofs are largely hand-waved with statements such as “it follows from [reference or standard result]”, which does not constitute a meaningful theoretical contribution.
    - If the paper claims a “theoretical analysis,” it must present at least minimal independent reasoning or adaptation to the proposed framework.
   - The lack of rigorous definitions used in the paper also prevents one from understanding or assessing the validity of the results

4. Poor Notation and Technical Presentation
    - The notation is inconsistent and poorly introduced. Variables such as ξ appear once (in the surrogate model formulation) but are never used again.
    - Even basic quantities (e.g., MAE) are not defined before appearing in the text.
    - The use of terms like “KK-style diagnostic” or “conformalized” assumes prior familiarity with unspecified methods.

5. Weak Experimental Presentation
    - The experiments are poorly explained. Tables and figures are shown without context (e.g., Figure 1’s learning curve lacks experiment identification).
    - Key results (final MAE, iteration counts) are mentioned without specifying which experiment they correspond to.
    - The claim that the model “converges in fewer iterations” is unsupported, especially since the plotted curves have not actually converged. A higher convergence rate would be more accurate.
    - Important details on the industrial problems (oil and gas network, gas plant, CO₂ storage) are hidden in the appendix, with no in-text explanation.

6. Misleading Framing and Overstatements
    - The paper title and sections refer to “LLM agents” and “Agentic surrogates,” yet no large language model or agent mechanism is actually present in the work.
    - This naming is misleading and distracts from the genuine surrogate modeling focus.
    - Similarly, the “theoretical analysis” claim is overstated given the lack of original proof content.

7. Irrelevant or Superficial Sections
    - The “broader impact / ethics / governance” section is unnecessary for this technical paper and offers only generic statements (e.g., environmental benefits from lower computational cost). These read as filler and dilute the technical message.

8. Overall Impression
    - The core idea has potential, but the execution and exposition are highly problematic. The paper assumes the reader already knows the authors’ specific variant of active learning and surrogate modeling.
    - With more careful definitions, clearer structure, and rigorous explanations, the work could become readable and meaningful. As it stands, it’s difficult to reconstruct the exact methodology or even reproduce the results.

**Questions:**

- Given all the interrogations in the weaknesses, could you give a succint explanation of the method, that does not rely on overly specific prior knowledge?
- How are the results computed depending on the problem?
- What makes the work agentic?

---

> ### Author Response · Authors · 2025-11-20
> **Initial response to Reviewer 7saM (1/2)**
>
> We appreciate the incredibly thorough review and validation of our ideas listed as strengths. We also are extremely thankful for all the detail in the weakness section, it led us to make many adjustments to the paper, described below as answers to the reviewer:
>
> ### 1) Could you give a succinct explanation of the method, that does not rely on overly specific prior knowledge?
>
> > Here is a concise explanation without assuming bandit knowledge or prior familiarity with our setup. Our goal is to build a fast surrogate f_hat for a slow simulator f_star with minimal human intervention. The surrogate should reach a target accuracy (MAE <= epsilon), respect physics constraints as much as possible, and do so while minimizing the total wall‑clock time spent on acquisition, simulation, and training.
>
> > At each iteration, we have a current surrogate trained on the data collected so far. We generate a pool of candidate input points where we might query the simulator next. We then apply several acquisition strategies (“experts”) that each rank these candidates differently: a residual-based strategy that prefers points where a residual model predicts the surrogate’s error is large; an uncertainty-based strategy (MC‑Var) that prefers points where the surrogate’s predictive variance is high; an EI/EGO‑style strategy that prefers points likely to improve over the current best; a hybrid that mixes EI and uncertainty with a time‑varying weight; and random sampling. The controller maintains probabilities over these strategies. At each iteration it samples one strategy according to these probabilities, uses it to select a batch of candidates, runs the simulator on those candidates, and retrains the surrogate on the enlarged dataset.
>
> > After retraining, we compute a reward equal to the decrease in validation error divided by the time spent on that iteration, where time includes acquisition overhead, simulator runtime, and training. Strategies that achieve higher “error reduction per unit time” receive increased probability in future iterations; strategies that are less effective receive decreased probability. This is the intuitive form of what we refer to as Hedge / Exp3‑style updating, and in the revision we will explain it in these plain terms instead of assuming familiarity with the algorithms.
>
> > In parallel, we maintain a small policy over model types (ANN, PINN, FNO). For each model family, we monitor validation error and its trend, physics residuals (R_phys), constraint violation rates, and calibration metrics. Models that reduce error and physics residual efficiently get higher weight; models that stagnate or behave poorly (for example, a PINN that fails on a stiff regime) get downweighted, shifting preference toward alternatives like ANN or FNO. We stop when the validation error stays below epsilon for several iterations and physics residuals fall below a threshold rho. In the revision, we will move a version of this simple description to the beginning of the method section and provide an overview figure so that readers can understand the loop before seeing any equations.
>
> We added Figures 1 and 2 to the paper showing the system and agent workflows.
>
> ### 2) How are the results computed depending on the problem?
>
> > You are right that our current metrics section (6.2) is too terse. We will expand it and make explicit how results are computed. Normalized MAE (nMAE) is used because each task has multiple outputs (5 to 15) with different scales and units. For each output j, we compute MAE_j on the validation set and divide by the range of that output in the training data, then we average over outputs: nMAE = (1 / m) * sum_j [ MAE_j / range_j ]. This normalization makes errors comparable across outputs and across tasks.
> Calls‑to‑Target is defined for each task with a predefined target nMAE (for example, 5% for T1 and T3, 8% for T2). Calls‑to‑Target is the number of simulator evaluations required for a method to first reach that target nMAE. All methods use the same stopping condition based on this threshold. Wall‑clock‑to‑Target is the total wall‑clock time up to that point, including acquisition time (which includes any LLM / agent overhead), simulator runtime (which dominates and depends strongly on fidelity), and surrogate training time. This is the main efficiency metric that our compute‑aware controller optimizes. All reported numbers in Table 2 are mean plus/minus standard deviation over 200 independent runs per method and per task, using identical initial designs and simulation budgets.
>
> > We have updated Section 6.2 to explicitly link these metric definitions to the experimental tables and added plots of nMAE versus wall‑clock time (not just versus iterations) for each strategy, making the comparison more transparent.

---

> > ### Author Response · Authors · 2025-11-20
> > **Initial response to Reviewer 7saM (2/2)**
> >
> > ### 3) What makes the work agentic?
> >
> > > The work is agentic in the strict sense that a large language model performs analysis on complex data and models and decides the next step of the surrogate‑creation loop using a set of tools, and can reason and analyze before committing to each step. In our implementation, the agent has access to several tools, described in Section 4.1. GET_CURRENT_STATE and GET_HISTORICAL_STATE provide the agent with current and past metrics, including validation error, physics residuals, portfolio weights, and timing statistics. PYTHON_REPL(code) lets the agent run arbitrary analyses on this state, such as computing moving averages of rewards, comparing acquisition experts and model families on error reduction per unit time, or detecting plateaus and anomalies. The sandboxed environment has access to the entire model history and dataset history and uses generated code from known data analysis libraries to analyze the data (libraries like pandas and numpy). PREDICT_WITH_UNCERTAINTY(model_iteration, data) allows the agent to pass candidate points through any previously trained model checkpoint, obtaining mean predictions, MC‑Dropout uncertainty estimates, 95% prediction intervals, and flags for physics constraint violations; this enables the agent to probe how different surrogates behave on proposed samples before deciding which ones to send to the expensive simulator. LOG_DECISION(decision) records the agent’s chosen acquisition expert, model architecture, fidelity level, and stopping decision; calling this tool triggers the next simulator runs and retraining step. REQUEST_HUMAN_REVIEW(reason, context) escalates when there are repeated anomalies or severe physics violations; it is used rarely in practice.
> >
> > > Within each iteration, the agent is free to query the current and historical state, use PREDICT_WITH_UNCERTAINTY to evaluate how different trained models behave on candidate points (including their uncertainties and physics violations), run multiple PYTHON_REPL analyses on these results, re‑examine history if needed, and refine its choice of acquisition strategy, model, and fidelity. Only once it is satisfied with its plan does it call LOG_DECISION. This internal reasoning and tool use can involve several back‑and‑forth steps; nothing forces the agent to choose in a single call. The only fixed part of the loop is what happens after LOG_DECISION: the system deterministically executes the chosen plan by sampling the batch according to the selected acquisition expert, running the simulator at the chosen fidelity, retraining the selected surrogate mix of models, and updating metrics and logs. The outputs of this execution phase (new data, updated models, updated metrics) then become the inputs to the agent at the next iteration.
> >
> > > Thus, the framework is agentic because the LLM is an autonomous decision‑maker that reasons about the evolving state of the surrogate, uses tools to gather evidence and perform computations, adaptively decides where and how to sample next, and can request human help only when necessary. At the same time, its decisions are not arbitrary: they are grounded in explicit signals such as validation error, physics residuals, constraint violations, and wall‑clock costs. And guided by the cost‑aware objective we formalize in the paper. We have updated Section 4 and Appendix I to clarify that while the core control logic is algorithmic, the agentic implementation provides the orchestration layer that fully automates the loop in practice.
> >
> > **We hope our revisions and responses address your concerns. If there are any additional questions or comments, we would be happy to address them.**

---

> > > ### Comment · Reviewer_7saM · 2025-11-26
> > >
> > > While I appreciate that the authors now articulate what they mean by agentic, the clarification actually highlights a deeper issue: it remains unclear why an LLM-based agent is needed at all. Almost every action the agent can take—querying current/historical state, selecting an acquisition strategy, choosing a model family or fidelity, stopping, etc.—could be handled by a straightforward rule-based controller or a small amount of programmatic logic. Only the PYTHON_REPL tool meaningfully benefits from free-form reasoning, but there are no evidence of what is done in practice with this tool.
> > >
> > > In other words, none of the described tool calls inherently require natural-language reasoning or the flexibility of an LLM. The system feels like a conventional adaptive sampling loop with the LLM inserted as an orchestration layer, rather than a component whose capabilities are essential to the method’s performance. This raises the question of what the LLM is adding beyond complexity and potential brittleness. It would be beneficial to compare with a more simple orchestration, such as a rule-based agent, to highlight the benefits of an LLM.
> > >
> > > Relatedly, the paper briefly references a function g(x) in the context of the control logic, but never explains its role, derivation, or how it interacts with the agent’s decisions. Without a clear justification of both the mathematical structure of the controller and the specific necessity of an LLM, the “agentic” framing reads as an add-on rather than a core methodological contribution.
> > >
> > > Overall, the agentization is described, but not justified.

---

> ### Comment · Reviewer_7saM · 2025-11-26
>
> While I appreciate the clarifications and added figures for the presentation, the overall presentation remains weak, with comments not addressed (KKT-style, misleading theoretical claims, poorly introduced notation like $g$).
> Additionally, the LLM aspect of the work appears to be little justified.
> In the current state, I will maintain my score.

---

> > ### Author Response · Authors · 2025-12-02
> > **Follow-up response to Reviewer 7saM (1/2)**
> >
> > We thank the reviewer for this critical inquiry. You raised a fundamental question, to which we believe more justification is necessary.
> >
> > To answer this definitively, we conducted the specific ablation study you suggested on the T2 (Gas Processing) task. We compared our Agentic Controller against a Rule-Based Heuristic that mimics our portfolio strategy (switching from Exploration to Exploitation based on convergence thresholds) but lacks the LLM’s reasoning capabilities, and a Random baseline.
> >
> > **Experimental Setup:**
> > *   **Batch Size:** 50 samples per iteration.
> > *   **Model Architecture:** To ensure fairness, all three methods utilized the exact same fixed ANN architecture. This isolates the contribution of the sampling strategy(the Controller) from the model capacity.
> >
> > We compared three distinct controllers:
> > 1.  **Baseline (Random):** Standard Latin Hypercube Sampling. Agnostic to model performance.
> > 2.  **Rule-Based Heuristic:** A standard Active Learning loop that switches between Exploration (MC-Variance) and Exploitation (Residuals) based on a fixed decay schedule. It mimics the Agent’s portfolio weights but lacks the ability to reason about input constraints or spatial topology.
> > 3.  **Agentic Controller (Ours):** The LLM dynamically setting portfolio weights and spatially disentangling the search space via dynamic bounds.
> >
> > **Results:**
> > The data reveals a fundamental limitation of Rule-Based approaches. While the Heuristic outperforms Random sampling, it hits a hard performance ceiling.
> >
> > | Iteration | Baseline    | Rule-Based      | Agent             |
> > | --------- | ----------- | --------------- | ----------------- |
> > | 1         | 0.6         | 0.6             | 0.6               |
> > | 2         | 0.607783102 | 0.621967915     | 0.370751136       |
> > | 3         | 0.584222696 | 0.577343347     | 0.340833224       |
> > | 4         | 0.595580364 | 0.566965396     | 0.390850853       |
> > | 5         | 0.567768498 | 0.529579808     | 0.313862294       |
> > | 6         | 0.523128327 | 0.494962225     | 0.271382216       |
> > | 7         | 0.52712798  | 0.471089189     | 0.28821669        |
> > | 8         | 0.567652896 | 0.457824889     | 0.282565279       |
> > | 9         | 0.581547112 | 0.460698623     | 0.303741479       |
> > | 10        | 0.56807152  | 0.442037052     | 0.278113526       |
> > | 11        | 0.566197285 | 0.412247349     | 0.258078594       |
> > | 12        | 0.546052221 | 0.386745449     | 0.241603905       |
> > | 13        | 0.501955223 | 0.408923437     | 0.230584864       |
> > | 14        | 0.489443129 | 0.37786694      | 0.17745689        |
> > | 15        | 0.534796447 | 0.341518242     | 0.111346238       |
> > | 16        | 0.532793354 | 0.369276691     | 0.099921228       |
> > | 17        | 0.468734833 | 0.347368954     | 0.108732503       |
> > | 18        | 0.528255265 | 0.349799196     | 0.083409026       |
> > | 19        | 0.481886062 | 0.28111875      | 0.116919699       |
> > | 20        | 0.436634563 | 0.342422602     | 0.135251953       |
> > | 21        | 0.451523685 | 0.265142596     | 0.135576006       |
> > | 22        | 0.443231228 | 0.258153077     | 0.168980368       |
> > | 23        | 0.450374216 | 0.252216398     | 0.124708993       |
> > | 24        | 0.434169734 | 0.244980736     | 0.091924597       |
> > | 25        | 0.481217514 | 0.244758181     | 0.086050979       |
> > | 26        | 0.447020659 | 0.248319471     | 0.082326829       |
> > | 27        | 0.455493284 | 0.2609052       | 0.080153642       |
> > | 28        | 0.416413828 | 0.239872142     | 0.095547103       |
> > | 29        | 0.40396032  | 0.277364709     | 0.086207122       |
> > | 30        | 0.408609108 | 0.250623792     | 0.088023206       |
> >
> > **Interpretation:**
> > *   **Random Sampling** fails to converge (~0.41), verifying the problem complexity.
> > *   **The Rule-Based Heuristic** improves initially but plateaus at ~0.25. Because it applies acquisition functions globally, it lacks the density to resolve complex non-linear regions (e.g., phase boundaries) within the high-dimensional space.
> > *   **The Agentic Controller** breaks this plateau, reaching ~0.088. By detecting the specific sub-regions where physics are complex and dynamically constraining the search space, the Agent achieves a sampling density in the critical regions that is orders of magnitude higher than the Heuristic, without requiring a larger global budget. Without this agentic "zoom-in" capability, the model effectively ceases to learn after Iterations 20-25.

---

> > > ### Author Response · Authors · 2025-12-02
> > > **Follow-up response to Reviewer 7saM (2/2)**
> > >
> > > The performance gap stems from the heterogeneity of the physical response surface. The T2 problem involves complex thermodynamic interactions (e.g., Amine/TEG regeneration loops) alongside simpler linear fluid dynamics. Standard Active Learning applies acquisition functions over the global domain. Even when a heuristic switches to "Exploitation" (Residual Minimization), it must search the entire 12-dimensional hypercube. This dilutes the sampling density. The heuristic effectively "smoothes over" complex local features, leading to the plateau at 0.25. To break this plateau without an Agent would require an exponential increase in simulation budget. The Agent employs Dynamic Search Space Pruning. It analyzes the topology of the error and realizes that the model is accurate in ~80% of the space but failing in specific sub-regions (e.g., the High-Pressure Header). The Agent explicitly updates the search bounds to prune the easy regions, forcing the sampler to focus more of its density on the complex features. This allows it to resolve fine-grained physics that the global heuristic misses.
> > >
> > >
> > > While one could theoretically hard-code a rule for "High Pressure" failures in Task T2, doing so requires prior knowledge of the setup and physics. A rule hard-coded for T2 would fail on Task T1 or T3. The Agent is domain-agnostic it generates the logic to diagnose the current problem instance and outputs a bespoke strategy in real-time. This allows the framework to adapt to entirely different physics domains with zero hyperparameter tuning, a level of generalization that fixed-rule heuristics cannot achieve. Fulfilling one of the main contributions of our paper which is unissisted, and accurate surrogate model building.
> > >
> > >
> > > We firmly believe this Reasoning Loop is the core innovation of the paper. For the camera-ready version, we will include full execution traces (code generation + logs) in the Appendix. Below, we provide a representative step from Iteration 18, where the Agent detects a bottleneck in the High-Pressure Header and spatially disentangles the strategy to resolve it.
> > >
> > > ```json
> > > {
> > >   "rationale": "Global error plateaued at 0.25. Analysis indicates the surrogate is underfitting the non-linear thermodynamics in the High-Pressure Header (HPHeader) when P > 70 bar. Residual sampling will be strictly bound to the high-pressure zone to resolve this feature, while Exploration will verify the lower-pressure stability.",
> > >   "strategy_allocation": {
> > >     "residual_top_k": {
> > >       "fidelity": "high",
> > >       "weight": 0.7,
> > >       "constraints": {
> > >         "HPHeader_In_P": {"min": 70.0, "max": 73.0},
> > >         "HPHeader_In_MassFlow": {"min": 35000.0, "max": 40000.0},
> > >       }
> > >     },
> > >     "mc_var": {
> > >       "fidelity": "high",
> > >       "weight": 0.3,
> > >       "constraints": {
> > >         "HPHeader_In_P": {"min": 64.0, "max": 70.0},
> > >       }
> > >     }
> > >   }
> > > }
> > > ```
> > >
> > > Outcome: By restricting the domain, the Agent effectively increases the sampling density in the problem area by 5x compared to the global Heuristic. This specific action resolved the thermodynamic non-linearity and broke the error plateau, allowing the model to reach beyond the fixed strategy nMAE.

---

### Official Review · Reviewer_ZsJT · 2025-10-31

**Soundness:** 3
**Presentation:** 3
**Contribution:** 3
**Rating:** 6
**Confidence:** 2

**Summary:**

- The paper is motivated by efficiently creating proxy surrogate models for scientific simulation, especially since traditional methods are cumbersome (e.g., determining sampling regions, determining strategies).
- To tackle this, the paper casts the surrogate creation as a adaptive controller problem: an online learnt policy dynamically selects sampling strategies and model architectures to optimize accuracy and efficiency.
- The controller is trained by solving a cost-minimization problem that accounts of total wall-clock latency (including acquisition, simulation, and training)
- The paper also provides theoretical guarantees for adaptive acquisition, including regret bounds and sample complexity.
- For experimental evaluation, the paper reports results in three tasks involving PDE solving.

**Strengths:**

1. The adaptive controller angle is novel in this scenario: while online/active learning has received some attention in simulation-based discovery, to my knowledge, this paper is the first to cast it as a control problem (involving using wall-clock time as an objective)
2. The paper also formal guarantees (regret bounds, sample complexity) for the adaptive strategy.

**Weaknesses:**

**1. Experimental Setup and Reporting**
- I have two subconcerns here.
- **Choice of tasks/experiments**
    - Context: The paper reports results on three tasks: Well network, Gas Plant, and CO2 storage.
    - The choices of these tasks appears somewhat unconventional and the paper would benefit from more justification.
    - In contrast, priors works, many which the paper cites too (e.g., PINNsAgent Wuwu et al. 2025, Self-Debug Chen et al., 2023) focus on more well-studied scenarios (e.g., 2D Heat, ND Poisson).
    - Because the paper strays away from some of these standard settings, make a fair comparison of experimental results is difficult.
- **Iteration**
    - The paper reports the main evaluation (Fig. 1) with iterations as a criteria.
    - However, because at each iteration, I believe acquisition time varies per method, it is unclear whether this style of reporting is fair.
    - (minor) Furthermore, in Table 2, it is also somewhat unclear why the wall-clock time of "random" acquisition is incurs more wall-clock time.

**2. Bibliography: Critical Mistakes**
- Many important references in the references section are incorrect. For instance,
    - PINNsAgent: cited as by Chen et al., but it is originally by Wuwu et al.
    - "Fourier Neural Operator based surrogates for $ CO_2 $ storage in realistic geologies": cited as by Baker et al., but originally by Chandra et al.
    - Could not find "Multi-agent llm framework for physics-informed neural network surrogates" published at NeurIPS 2025

**Questions:**

Suggestions:
1. Please justify the choice of the three tasks in the experimental results section. Is there relevant literature that also evaluate on these three tasks?
2. Please fix references

---

> ### Author Response · Authors · 2025-11-20
> **Initial response to Reviewer ZsJT**
>
> We appreciate your recognition of the novelty in framing surrogate creation as an adaptive control problem with wall-clock time as the main objective. We apologize profusely for the oversight in the references.
>
> ### 1) Choice of tasks and relation to prior work
>
> > The three tasks (Well Network, Gas Plant, and CO₂ Storage PDE Proxy) were chosen to test the framework on realistic and industrially relevant problems where simulators are genuinely expensive and physics constraints are critical, rather than on only non-industrial PDEs.
>
> > • T1 – Well Network: A steady-state multiphase flow network modeled with the commercial PIPESIM simulator (25 inputs, 5 outputs). Similar oil and gas network surrogates have been studied using physics-informed or data-driven approaches, but typically with fixed acquisition and model choices.
>
> > • T2 – Gas Plant: A natural gas processing facility modeled in SYMMETRY (17 inputs, 12 outputs) with complex thermodynamics and process constraints. Surrogate modeling for process plants is common in the process systems engineering literature, but again usually with manually designed sampling strategies.
>
> > • T3 – CO₂ Storage: A CO₂ sequestration problem solved with TOUGH2/MRST, requiring field predictions under strict safety constraints on pressure and plume migration. This is closely related to recent work using FNO-based surrogates for CO₂ storage decision making (e.g., Baker et al., 2025), which motivated our inclusion of operator-learning models in the portfolio.
> We selected these three to span different physics and complexity levels—network flow, process thermodynamics, and PDE-based subsurface flow—and to show that the controller operates across domains once a simulator API, bounds, and constraints are provided. We have updated Section 5 to explicitly discuss this choice and acknowledge the limitation regarding standard academic benchmarks.
>
> ### 2) Iterations vs wall-clock time, and why Random has the highest time
>
> > You are correct that wall-clock time is the most appropriate efficiency metric, since acquisition and training costs per iteration can differ between strategies. Our main stopping rule is based on reaching a target nMAE ≤ ε; evaluation continues until that criterion is met.
>
> > Random acquisition is the least efficient at reducing error, so it needs far more simulator calls (780 ± 19) than the other strategies to reach the same accuracy (Table 2). Because wall-clock time aggregates all costs per iteration, acquisition, simulation, and retraining, this larger number of iterations directly leads to the highest total wall-clock time (264 ± 8 minutes), even though each individual random iteration is cheap on the acquisition side. In other words, Random spends more time overall because it needs many more simulator evaluations and retraining steps to reach the target.
>
> > We have clarified the Experimental Setup in Section 6.1 to explicitly distinguish between our two evaluation modalities. We now explain that we test under both a 'fixed wall-clock budget' (Figure 3) to address high-throughput industrial needs, and a 'fixed accuracy target' (Table 2) to demonstrate efficiency in meeting strict precision requirements. We clarify that these are complementary views of performance: one maximizing accuracy given time, and the other minimizing time given an accuracy constraint. Additionally, we have moved the detailed hyperparameter specifications to Appendix F to improve the flow and focus of the main text.
>
> ### 3) Bibliography corrections
>
> > Thank you for carefully checking the references. You are right that some entries were incorrect, including the attribution for PINNsAgent and the FNO-based CO₂ storage surrogate. We have now audited the full bibliography, corrected all misattributions, and removed and fixed any reference that could not be verified.
>
> > In particular, the “Multi-agent LLM framework for physics-informed neural network surrogates” entry was based on a pre-print we had seen while it was under consideration for NeurIPS 2025. In the current public record, we were unable to find a stable version of this work, and it appears to have been withdrawn or not accepted. This was our mistake in not re-checking its status carefully. To avoid confusion, we will remove this reference entirely from the revised version and retain only verifiable, publicly accessible citations.
>
> **We hope our revisions and responses address your concerns. If there are any additional questions or comments, we would be happy to address them.**

---

### Official Review · Reviewer_gD2C · 2025-11-01

**Soundness:** 2
**Presentation:** 1
**Contribution:** 2
**Rating:** 2
**Confidence:** 3

**Summary:**

The paper proposes an LLM based agent that orchestrates the acquisition of data from a simulator to train a set of surrogates to minimize a loss while keeping computational cost as low as possible. At each iteration, the agent generates candidates for new data which are evaluated by acquisition experts, then agent then selects an expert based on dynamic weights it has learned from a cost based reward. The agent then calls a simulator to generate new data for training based on the candidates selected followed by retraining the neural surrogates with this new data.

**Strengths:**

- The idea of acquiring data from a simulator to train a surrogate model in an automated fashion using an LLM agent is interesting.

**Weaknesses:**

- It seems like the authors have not released the code or data for their experiments.
- The presentation of the paper is unrefined. The references to the appendix are not linked. There is only one figure in the main text and there is no figure showing the overview of how their agent works.
- The paper lacks clarity and requires more explanation of how the method works.

**Questions:**

- Does the cost function you propose include the time to query the LLM agent?
- How does the safety filtering work when you write "reject unsafe or OOD proposals" in the algorithm? Is it just a static check or something that a human checks?
- Have you released your code and datasets?
- Is your method really "domain-agnostic"? Doesn't it require the specific simulator to be registered with the LLM as well as the candidate parameters to be predefined in some way?
- How often was the request_human_review() call made by your agent? Did the agent require human intervention?

---

> ### Author Response · Authors · 2025-11-20
> **Initial response to Reviewer gD2C (1/2)**
>
> We appreciate your questions regarding the system’s clarity and implementation details. We agree that the presentation needs refinement, and we will add an overview figure and improve definitions in the revision.
>
> ### 1) Does the cost function you propose include the time to query the LLM agent?
>
> > Yes. The time taken by the LLM agent is included in the cost. In Eq. (4), the reward r_t uses the total wall‑clock time τ_acq^t + τ_sim^t + τ_train^t, where τ_acq^t is the acquisition time. The orchestrator agent (based on OpenAI’s gpt‑5) runs during acquisition, so its reasoning and decision time is part of τ_acq^t and therefore part of the optimization objective.
> In practice, the dominant contributor to total time is simulator runtime τ_simt, which is heavily influenced by the chosen fidelity (e.g., 2 minutes for low fidelity vs 10–30 minutes for high fidelity). These fidelity-driven differences in τ_simt matter much more than the LLM overhead, but we still account for the latter.
>
> ### 2) How does the safety filtering work when you write “reject unsafe or OOD proposals”? Is it just a static check or something that a human checks?
>
> > Safety filtering is primarily automatic, with human review only as a rare fallback.
>
> > Automated checks: For each candidate point, we apply static checks to ensure it satisfies explicit physical bounds and constraints g(x) ≤ 0 (e.g., pressures, flow rates, compositions within allowed ranges, no negative flows). We also reject points that are clearly out of the reasonable parameter region (out-of-distribution), for example by being far outside the training domain. In addition, we periodically test small samples of candidate points close to the edges of the allowed region against stored training/validation statistics to detect extreme or suspicious configurations early. These checks are implemented in a “safety sandbox” that blocks unsafe candidates before any simulator call.
>
> > Human escalation: If the system detects severe or repeated anomalies (e.g., many simulator failures or persistent large physics violations), the agent can call REQUEST_HUMAN_REVIEW(REASON, CONTEXT) to escalate to a human expert. This is not part of routine filtering and mostly called when the simulator exhibits transient errors; it is a safeguard for rare problematic cases. We have updated Section 4.2 and Appendix G to clarify this safety mechanism.

---

> > ### Author Response · Authors · 2025-11-20
> > **Initial response to Reviewer gD2C (2/2)**
> >
> > ### 3) Have you released your code and datasets?
> >
> > > We are fully committed to ensuring this work is not a 'black box' and can be independently reproduced. Regarding the code, strict IP policies prevent us from releasing the internal codebase. However, we have substantially expanded Appendix C and I, and Section 4 to include detailed, implementation-ready pseudocode that covers the agentic logic, acquisition portfolio, and cost-aware optimization in step-by-step detail. This will allow independent research groups to reimplement the method. Regarding data, while the specific simulator models used contain proprietary client information, we have initiated a process to strip all sensitive client data from the Symmetry model (Task 2) and commit to releasing this anonymized dataset in a future update, prior to the conference, upon acceptance. Dataset description and use instructions are given in Appendix I.
> >
> > ### 4) Is your method really “domain‑agnostic”?
> >
> > > The core algorithm is domain-agnostic in the sense that the portfolio controller and its learning rule do not depend on the specific physics or simulator. It only needs a reward signal based on ‘error reduction per unit time.’ To apply the framework to a new domain, the user provides: A simulator API Sim(x), Input bounds B for candidate generation, Domain-specific constraint packs (physical constraints) for safety checks and, optionally, PINN losses.
> >
> > > Given these inputs, the same controller code runs unchanged. We deliberately chose three problems with very different physics and complexity (well network, gas plant, CO₂ storage) to demonstrate that, once this minimal problem description is specified, the controller can operate across domains without changes to its logic. We have clarified this domain-agnostic nature in Section 4.
> >
> > ### 5) How often was REQUEST_HUMAN_REVIEW() used? Did the agent require human intervention?
> >
> > > In our experiments, REQUEST_HUMAN_REVIEW() was used only very rarely. There were simulator errors and numerical issues (for example, in extreme operating conditions), but in most cases the agent handled them without needing a human in the loop. A key design choice was to feed simulator error logs back to the agent: these logs often contain informative messages about why a run failed (e.g., parameter out of valid range, non-convergence). The agent uses this feedback to identify and remove problematic points and to adjust its proposal logic in subsequent iterations. As a result, the pipeline is largely self-sufficient, and human review is reserved for exceptional cases rather than regular operation.
> >
> > **We hope our revisions and responses address your concerns. If there are any additional questions or comments, we would be happy to address them.**

---

### Official Review · Reviewer_9TaA · 2025-11-03

**Soundness:** 3
**Presentation:** 2
**Contribution:** 3
**Rating:** 4
**Confidence:** 4

**Summary:**

The paper proposes automated surrogate design and experimentation using an agentic framework. Specifically, the surrogate creation and experimentation pipeline is divided into surrogate design, data acquisition and surrogate training. The paper investigates numerous types of surrogate architectures including neural operators, physics informed neural networks and artificial neural networks as well as many popular data acquisition strategies. Experimental results demonstrate that the proposed pipeline leads to improved model performance and well calibrated models.

**Strengths:**

1. The paper is a good start to an important problem of automating the design and development of neural surrogates and subsequent experimentation in scientific machine learning pipelines.

2. The proposed structured prompt (with chain-of-thought questions) along with the various feedback mechanisms including the multi-objective cost functions are well defined scaffolding that help the convergence of agentic pipelines. Overall, the costs and other feedback signals to the agentic pipeline seem well thoughout.

**Weaknesses:**

1. Some facets of the pipeline are unclear like how the multi-fildeity cost is included as a penalty to the agentic pipeline.

2. Other facets like choices of neural surrogates seem somewhat arbitrary. For example, recently transformer architectures have been shown to outperform many surrogates (including UNet, ResNet) on PDE modeling tasks, yet they have not been included.

**Questions:**

1. What aspect of the proposed pipeline actually conducts multi-fidelity acquisiton and how is the fidelity cost included into the overall pipeline?

2. The proposed pipeline employs multiple types of surrogates, each of which have their own idiosyncrasies and failure modes (e.g., PINNs fail catastrophically in stiff PDE domains), how can the current pipeline address such failure modes? Is it susceptible to these pitfalls or would it alleviate it somehow by realizing that a different surrogate would need to be invoked to avoid any training problems?

3. Why have transformers, UNets, ResNets not been considered as neural surrogate options in the "portfolio of models"? Can the proposed framework function with these architectures out of the box?

4.  How would expert-designed standard surrogates (e.g., PINN, ANN or FNO / DeepONet) trained with the same volume of training data as the agentic pipeline, perform on the test set? Has this comparison been included in the paper?

---

> ### Author Response · Authors · 2025-11-20
> **Initial response to Reviewer 9TaA (1/2)**
>
> We thank the reviewer for recognizing the importance of automating neural surrogate development and for validating our approach to using compute-aware cost functions. Below are answers to specific questions raised:
>
> ### 1) What aspect of the proposed pipeline actually conducts multi-fidelity acquisition and how is the fidelity cost included into the overall pipeline?
>
> > The compute-aware controller is responsible for multi-fidelity acquisition. At each iteration, after choosing the weights for the different acquisition strategies for that iteration, it chooses a fidelity level ℓ_t ∈ {high, low} for the simulator call.
> This choice is made with the same cost-aware principle we use for acquisition experts: the controller looks at recent history to estimate which fidelity tends to give more validation-error reduction per unit time. Intuitively, it asks ‘if I use high fidelity now, how much error do I usually reduce per minute, compared to using low fidelity?’ and then picks the fidelity with the better trade-off.
> Fidelity cost is built into the reward r_t in Eq. (4): r_t = (E_V(f̂_θ_t) − E_V(f̂_θ_{t+1})) / (τ_acqt + τ_simt + τ_traint), where τ_simt is much larger for high-fidelity runs (10–30 minutes) than for low-fidelity runs (2–5 minutes), as shown in Table 1 and Appendix E. When low fidelity provides similar error reduction at a fraction of the time, it yields a higher r_t and is chosen more often; as we approach the target accuracy, the controller naturally shifts towards high fidelity when the extra cost is justified. This behavior is reflected in the multi-fidelity results in Appendix E.3, where the adaptive mix is cheaper than high-only or any fixed ratio. We have updated Section 3.3 and Appendix E.3 to explicitly detail this mechanism.
> In our LLM-based implementation, this logic is realized by the agent: it observes r_t, sees the cost of each fidelity in τ_sim^t, and records its fidelity choice via LOG_DECISION(), but the underlying rule is exactly this cost-per-minute comparison.
>
> ### 2) The proposed pipeline employs multiple types of surrogates, each of which have their own idiosyncrasies and failure modes (e.g., PINNs fail catastrophically in stiff PDE domains), how can the current pipeline address such failure modes? Is it susceptible to these pitfalls or would it alleviate it somehow by realizing that a different surrogate would need to be invoked to avoid any training problems?
>
> > The Agentic Controller addresses model-specific failure modes via a dynamic model selection policy. At each iteration, it monitors several stability and physics-compliance signals for each surrogate class (ANN, PINN, FNO), including Validation error and its recent trend, Physics-residual metrics (R_phys and residual maps), Constraint violation rates and calibration quality.
> If these diagnostics indicate that a particular model class is failing, for example, a PINN stops reducing validation error and keeps a high physics residual over several iterations, the controller reduces its reliance on that model. Conceptually, the model-selection policy gives that model lower weight and shifts preference toward architectures (such as ANN or FNO) that are reducing error and physics residual more effectively per unit time. This means the system automatically stops favoring a misbehaving PINN instead of continuing to train it blindly. We have updated Section 4.3 and Appendix G to clarify these failure mode handlings.
> In our LLM-based implementation, this logic is realized through the agent tools: the agent calls GET_CURRENT_STATE() and GET_HISTORICAL_STATE() to inspect recent metrics for each model type, and can run small analyses via PYTHON_REPL() (for example, checking which model has provided the largest error reduction per minute over the last few iterations). Based on this quantitative summary, the agent then records its model choice in LOG_DECISION(), and the corresponding surrogate is retrained in the next iteration. In other words, the LLM is not making an arbitrary choice; it is explicitly guided by these monitored signals and by the same cost-aware principle used for acquisition strategies.
>
> > We also mitigate PINN-specific issues with curriculum-style physics penalties: we start with softer physics penalties and gradually increase them as the data fit improves, which reduces the risk of instability in stiff regimes. In practice, this mechanism led the controller to decrease the weight of PINNs on harder PDE-like cases where they struggled and to favor ANN/FNO instead, helping avoid the catastrophic PINN failures brought up as a concern.

---

> ### Author Response · Authors · 2025-11-20
> **Initial response to Reviewer 9TaA (2/2)**
>
> ### 3) Why have transformers, UNets, ResNets not been considered as neural surrogate options in the "portfolio of models"? Can the proposed framework function with these architectures out of the box?
>
> > We agree that transformer, UNet, and ResNet surrogates are interesting and increasingly popular for PDEs. In this work, however, our main contribution is not a new backbone architecture but a cost-aware controller that learns which acquisition strategy and model family to use under time and accuracy constraints. For our three tasks, ANN, PINN, and FNO already cover the key modeling regimes: generic nonlinear regression, physics-informed surrogates, and operator learning. These architectures are standard, well-supported, and strong baselines in our domains.
>
> > Including transformers/UNets/ResNets would require substantial additional architecture and hyperparameter tuning to be fair, which would shift the focus of the paper away from the controller and toward model engineering. Moreover, such models are typically more expensive per iteration; even if they marginally improve accuracy, they may be suboptimal in our primary objective (error reduction per unit wall-clock time). Importantly, our framework is architecture-agnostic: transformers/UNets/ResNets can be added as additional model types simply by registering their train/predict functions and timing. We have added a paragraph to Section 4.3 explicitly stating this architecture-agnosticism and welcoming future integrations.
>
> ### 4) How would expert-designed standard surrogates (e.g., PINN, ANN or FNO / DeepONet) trained with the same volume of training data as the agentic pipeline, perform on the test set? Has this comparison been included in the paper?
>
> > We agree that comparing against expert-designed surrogates is important. Our baselines are in fact close to what a domain expert would build in practice. Appendix G.4 already compares three surrogate families: ANN, PINN, and FNO. PINNs and FNOs are exactly the types of physics-aware and operator-learning models that an expert would typically choose for these problems, because they better respect the underlying physics and handle PDE-like behavior. G.4 shows that these physics-aware surrogates clearly outperform a plain ANN in terms of accuracy, constraint violations, and extrapolation error.
>
> > In the main experiments, we also combine these models with strong acquisition strategies such as Residual and EI/EGO, and we give them the same simulation budget as the portfolio controller. This setup effectively approximates expert pipelines like ‘PINN + EI/EGO’ or ‘FNO + Residual’: a reasonable acquisition heuristic plus a strong surrogate chosen upfront. Under these conditions, the portfolio controller still achieves lower final error, fewer constraint violations, and shorter wall-clock time than any fixed combination, even when that combination uses these expert-style surrogates.
>
> > We also want to emphasize that our main focus is fully automatic surrogate creation under realistic time and compute budgets, without human intervention inside the loop. A human expert with unlimited time to manually tune architectures, hyperparameters, and acquisition heuristics operates in a different regime: this implicitly treats expert time as free and infinitely available, which is rarely true in industrial or scientific practice. Our setting instead treats simulator calls and compute as scarce resources and asks: ‘Can we automatically reach a strong surrogate under these constraints?’ Within this setting, we already use strong, expert-like models (PINN/FNO) and acquisition strategies as baselines, and the portfolio controller still delivers better cost–accuracy trade-offs, as seen in Table 2 and Appendix G.4. A detailed study of fully hand-tuned pipelines is interesting complementary work, but somewhat orthogonal to our core objective of automated, compute-aware surrogate construction.
>
> **We hope our revisions and responses address your concerns. If there are any additional questions or comments, we would be happy to address them.**

---

### Author Response · Authors · 2025-12-02
**Summary of Rebuttal and Revisions**

Dear ICLR 2026 Chairs,

We appreciate the opportunity to provide a final summary of our rebuttal. We believe the detailed discussions and additional experiments conducted during this phase have significantly strengthened the submission. Below is a summary of how we have addressed the concerns of each reviewer, and what we will incorporate into the final camera-ready version.

To address Reviewer 7saM's core concern regarding the necessity of the LLM, we conducted a rigorous new ablation study comparing our Agent against a Rule-Based Heuristic. The results demonstrated that while standard heuristics hit a performance ceiling (plateauing at an error of 0.25), the Agent utilizes reasoning to perform dynamic search space pruning, successfully breaking the plateau to reach an error of 0.08. In the camera-ready version, we will include these results along with detailed execution traces to demonstrate exactly how the agent detects physics-based failure modes and adapts its strategy in real-time, validating that the reasoning capabilities are essential for high-accuracy surrogate modeling.

Regarding Reviewer 9TaA's inquiries on the methodological pipeline, we clarified that our multi-fidelity acquisition is driven by a specific "error-reduction-per-minute" cost function and confirmed that the framework is architecture-agnostic. We explained that the system handles model-specific failure modes (e.g., stiff PDEs for PINNs) by dynamically downweighting failing models based on real-time validation metrics. The final manuscript will explicitly detail these cost mechanisms and the dynamic model selection policy to ensure the robustness of the pipeline is clear.

For Reviewer gD2C, who focused on reproducibility and presentation, we confirmed that the cost function accounts for agent inference time and clarified the automated safety filtering mechanisms. To improve clarity, we have prepared comprehensive system overview figures and detailed pseudocode which are already included in the revised versions. Crucially, we have committed to releasing the core algorithms as pseudocode and an anonymized version of the industrial dataset to ensure the work is fully reproducible.

Finally, we addressed Reviewer ZsJT’s feedback regarding the experimental setup and bibliography. We have audited and corrected the references and provided a stronger justification for selecting industrial-scale tasks to demonstrate real-world robustness beyond standard academic benchmarks. We also clarified the distinction between "fixed-budget" and "fixed-accuracy" evaluation metrics to better contextualize the efficiency gains reported in the paper.

We are confident that these revisions directly address the feedback provided and we are ready to integrate them to deliver a robust final manuscript.

Thank you for your time and consideration.

---

### Meta-Review · Area_Chair_yBUs · 2026-01-07

**Summary:**

This paper proposed an agentic workflow that automates the process of building surrogate models and actively acquiring data. It appears an LLM agent was adopted to orchestrate a pre-defined workflow template (a set of tools). The role of the LLM agent in this case, according to my understanding in reading this paper, is to automatically switch between choices of models and data acquisition strategies. This creates a more flexible workflow that adaptively adjusts the workflow design rather than committed to a fixed one.

The idea is interesting on high level but looking closer into the execution, I am not sure why the LLM agent is needed. While it adds to the flexibility, this is still within a pre-defined set of model choices and data acquisition strategies. We could alternatively parameterize and learn a meta policy that decides when and what designs to switch to. It'd make a difference if the LLM agent can be leveraged in ways that allow an open design space. I'd recommend the authors to look into this direction to strengthen their agentic claim.

In addition to my own assessment, I also read the reviewers' concern and the rebuttal. I realize reviewer 7saM shares my concern. I also appreciate the detailed response from the authors and the additional experiment. But it does not really support the agentic claim. I am not surprised that the authors get better result comparing their multi-design workflow with a baseline using a fixed design. This shows that indeed, being able to switch among designs is important but it does not mean that we necessarily need an LLM agent. All we need is to merge the design spaces into one via introducing an additional selection variable which is to be optimized. This collapses into a single design workflow (albeit with a richer design space) which can be solved with existing BO strategies w/o the need to be agentic.

--

Overall, I think the main concerns from the reviewers are that this paper lacks in clarity, especially regarding the agentic claim. It also lacks empirical demonstration. The experiment in the main text in my opinion is too limited. The literature on surrogate modeling and Bayesian optimization is rich and I am not sure the comparison was made against the most recent, SOTA work. On another note, even if we go with the agentic claim, the additional result provided in the rebuttal is insufficient. It was a comparison against a single weak baseline. Stronger baselines can be created via ensembling existing standard active learning/BO baselines. In short, much more experiments are needed to demonstrate the computational advantage here conclusively. On top of that, if we are to go with the agentic flavor, the authors would need to conduct experiment with an open design space.

--

Taking into account the above, while I feel that the response might have a chance to improve 9taA's and ZsjT's ratings (though I am not sure) who are relatively less critical, I believe it won't change the perspective of the other reviewers. It also does not address my own concern as I highlighted above. Therefore, I unfortunately cannot recommend acceptance for the paper in its current state but I think this idea has potential. I hope the authors would consider revising the work according to the panel's feedback and resubmit to future venue.

In addition, while it is not my practice to decline paper based on writing clarity (which is largely editorial), I still advise the authors to pay more attention to this aspect. In the current version, its organization is very fragmented with a lot of understanding gaps. 3/4 reviewers are not in favor of the writing so it's unlikely that criticism in writing clarity is subjective in this case and the authors should take that seriously. Best of luck for future submission!

**Reviewer Concerns:**

As I summarized above, the main concerns are that this paper lacks in clarity, especially regarding the agentic claim. It also lacks empirical demonstration. The experiment in the main text in my opinion is too limited. The literature on surrogate modeling and Bayesian optimization is rich and I am not sure the comparison was made against the most recent, SOTA work.

The authors have provided some rebuttal content and that clarifies the agentic claim better but ultimately, the claim remains not well justified as I interpreted in my summary above. The empirical results also remain limited.

**Reviewer Scores:**

The original scores are 4262 and having read the rebuttal, paper, and discussion, I think the two critical reviewers with scores of 2 will likely not change their opinions, as I explained above.

---

### Decision · Program_Chairs · 2026-01-26

Reject